



# Actors, actions, and uncertainties: optimizing decision-making based on 3-D structural geological models

**Fabian Antonio Stamm**[1], **Miguel de la Varga**[1,2], **and Florian Wellmann**[1]

[1]Computational Geoscience and Reservoir Engineering (CGRE), RWTH Aachen University, Germany
[2]Aachen Institute for Advanced Study in Computational Engineering Science (AICES), RWTH Aachen University, Germany

**Correspondence:** Fabian Antonio Stamm (fabian.stamm@rwth-aachen.de)

**Abstract.** [TS1 CE1]Uncertainties are common in geological models and have a considerable impact on model interpretations and subsequent decision-making. This is of particular significance for high-risk, high-reward sectors. Recent advances allows us to view geological modeling as a statistical problem that we can address with probabilistic methods. Using stochastic simulations and Bayesian inference, uncertainties can be quantified and reduced by incorporating additional geological information. In this work, we propose custom loss functions as a decision-making tool that builds upon such probabilistic approaches.

As an example, we devise a case in which the decision problem is one of estimating the uncertain economic value of a potential fluid reservoir. For subsequent true value estimation, we design a case-specific loss function to reflect not only the decision-making environment, but also the preferences of differently risk-inclined [CE2] decision makers. Based on this function, optimizing for expected loss returns an actor's best estimate to base decision-making on, given a probability distribution for the uncertain parameter of interest. We apply the customized loss function in the context of a case study featuring a synthetic 3-D structural geological model. A set of probability distributions for the maximum trap volume as the parameter of interest are generated via stochastic simulations. These represent different information scenarios to test the loss function approach for decision-making.

Our results show that the optimizing estimators shift according to the characteristics of the underlying distribution. While overall variation leads to separation, risk-averse and risk-friendly decisions converge in the decision space and decrease in expected loss given narrower distributions. We thus consider the degree of decision convergence to be a mea-

sure for the state of knowledge and its inherent uncertainty at the moment of decision-making. This decisive uncertainty does not change in alignment with model uncertainty but depends on alterations of critical parameters and respective interdependencies, in particular relating to seal reliability. Additionally, actors are affected differently by adding new information to the model, depending on their risk affinity. It is therefore important to identify the model parameters that are most influential for the final decision in order to optimize the decision-making process.

## 1 Introduction

In studies of the subsurface, data availability is often limited and characterized by high possibilities of error due to signal noise or inaccuracies. This, together with the inherent epistemic uncertainty of the modes, leads to the inevitable presence of significant uncertainty in geological models, which in turn may affect interpretations and conclusions drawn from a model (Wellmann et al., 2018, 2010a; de la Varga and Wellmann, 2016; de la Varga et al., 2019; Bardossy and Fodor, 2004; Randle et al., 2019; Lark et al., 2013; Caers, 2011; Chatfield, 1995). Uncertainties are thus of particular importance for making responsible and good decisions in related economic settings, such as in hydrocarbon exploration and production (Thore et al., 2002; McLane et al., 2008; Smalley et al., 2008). The quantification and visualization of such uncertainties and their consequences is currently an active field of research. Recent developments allow us to view geological modeling as a statistical problem (see Wellmann and Caumon, 2018). We particularly regard approaches to couple

implicit geological modeling with probabilistic methods, as presented by de la Varga et al. (2019) with the Python library GemPy.

Building on this probabilistic perspective, we propose the use of custom loss functions as a decision-making tool when dealing with uncertain geological models. In many applications, we are interested in some decisive model output value, for example reservoir volume. Given that such a parameter is the result of a deterministic function of uncertain variables in our model, the parameter of interest is likewise uncertain and can be represented by a probability distribution attained from stochastic simulations. A loss function can be applied to such a distribution to return a case-specific best estimate to base decision-making on.

We consider hydrocarbon exploration and production as an exemplary high-risk, high-reward sector, in which good decision-making is crucial. However, the described methods are potentially equally applicable to other types of fluid reservoirs (e.g., groundwater geothermal or $CO_2$ sequestration) and in the raw materials sector. Monte Carlo simulation for reservoir estimation and risk assessment has become common in this sector and is often used in combination with decision trees (see Murtha et al., 1997; Mudford et al., 2000; Wim et al., 2001; Bratvold and Begg, 2010). However, it seems to us that distributions resulting from probabilistic modeling are mostly only considered to attain best estimates in the form of means. The most likely and extreme outcomes are identified as percentiles, typically P50 (the median), P10, and P90. We believe that this practice does not harness the full potential of such a probabilistic distribution and that much of the inherent information is discarded. Contrary to that, customized loss functions, as a Bayesian method, take into account the full probability distribution and enable the inclusion of various conditions in the process of finding an optimal estimate. While used in statistical decision theory and other scientific fields, loss functions have, to the best of our knowledge, found no significant application in the context of structural geological modeling. Thus, we intend to provide a new perspective with our methodology.

To illustrate our approach of using custom loss functions for decision-making, we first illustrate what such customization might look like step by step: starting off with a standard symmetrical loss function, incorporating scenario-specific conditions and assumptions, and lastly implementing a factor to represent the varying risk affinities of different decision makers. As we assume a petroleum exploration and production decision-making scenario, our parameter of interest should be one that indicates the economic value of a potential hydrocarbon accumulation. In a larger context, including various geological and economic factors such as operational expenditures, this could be the net present value (NPV) of a project. In preproduction stages, original oil in place (OOIP) is commonly used for early assessments (Dean, 2007; Morton-Thompson et al., 1993). Decision makers would want to best estimate the relevant parameter of interest to derive recoverable reserves, create economic value, and subsequently allocate development resources accordingly, which includes the possibility of walking away from a prospect. In this case, the decision maker might refer to an individual geological expert, but also to an exploration company as a whole.

Once we have set up a loss function customized to this decision problem, we can apply it to probability density functions that represent our knowledge about the true value of the parameter of interest. As mentioned above, such distributions can result from geological modeling in a probabilistic context. To illustrate this, we include a synthetic 3-D structural geological model as a case study. In this context, we define the structurally determined maximum trap volume $V_t$ as our parameter of interest and indicator for economic value. We generate different probability distributions via stochastic simulations and based on various information scenarios. It is important to note that these are always based on the same primary input parameters. We attain altered states of information by updating the reference case (prior) with secondary information. In doing so, we make sure that the resulting distributions and the realizations of loss function applications can be directly compared. These case studies are synthetic and chosen here to exemplify the application of Bayesian decision theory and to show how additional information affects the optimality of decision.

## 2 Methods

### 2.1 Bayesian decision theory

We view the statistical analysis of geological models from a probabilistic perspective, which is most importantly characterized by its preservation of uncertainty. Its principles have been presented and discussed extensively in the literature (see Jaynes, 2003; Box and Tiao, 2011; Harney, 2013; Gelman et al., 2014; Davidson-Pilon, 2015). The Bayesian approach is widely seen as intuitive and inherent in the natural human perspective. It regards probability as a measure of belief about a true state of nature.

In many cases, decisions are made on the basis of summary parameters such as mean or standard deviation. This approximation works for well-defined probability distributions but it may fail when the distribution does not have a defined structure, which is the usual case of distribution generated as a result of a Bayesian inference. In this work, we aim to tackle the decision problem associated with probabilistic inferences. By applying Bayesian decision theory concepts, we are capable of transforming an arbitrary complex set of distributions onto a more adequate dimension for decision-making, usually loss or score.

### 2.1.1 Loss, expected loss, and loss functions

Common point estimates, such as the mean and the median of a distribution, usually come with a measure for their accuracy (Berger, 2013). However, it has been argued by Davidson-Pilon (2015) that using pure accuracy metrics, while this technique is objective, ignores the original intention of conducting the statistical inference in cases in which payoffs of decisions are valued more than their accuracies. A more appropriate approach can be seen in the use of loss functions (Davidson-Pilon, 2015).

Loss is a statistical measure of how "bad" an estimate is. Estimate-based decisions are also referred to as actions $a$. Therefore, we also refer to decision makers as actors. Loss is defined as $L(\theta, a)$, so $L(\theta_1, a_1)$ is the actual loss incurred when action $a_1$ is taken, while the true state of nature is $\theta_1$ (Berger, 2013). The magnitude of incurred loss related to an estimate is defined by a loss function, which is a function of the estimate and the true value of the parameter (Wald, 1950; Davidson-Pilon, 2015):

$$L(\theta, \hat{\theta}) = f(\theta, \hat{\theta}). \tag{1}$$

So, how "bad" a current estimate is depends on the way a loss function weights accuracy errors and returns respective losses. Two standard loss functions are the absolute-error and the squared-error loss function. Both are objective, symmetric, simple to understand, and commonly used.

The presence of uncertainty during decision-making implies that the true parameter value is unknown and thus the truly incurred loss $L(\theta, a)$ cannot be known at the time of making the decision. The Bayesian perspective considers unknown parameters to be random variables and samples that are drawn from a probability distribution to be possible realizations of the unknown parameter; i.e., all possible true values are represented by this distribution.

Under uncertainty, the expected loss of choosing an estimate $\hat{\theta}$ over the true parameter value $\theta$ is defined by (Davidson-Pilon, 2015)

$$l(\hat{\theta}) = E_\theta[L(\theta, \hat{\theta})]. \tag{2}$$

The expectation symbol $E$ is subscripted with $\theta$, by which it is indicated that $\theta$ is the respective unknown variable. This expected loss $l$ is also referred to as the Bayes risk of estimate $\hat{\theta}$ (Berger, 2013; Davidson-Pilon, 2015).

By the law of large numbers, the expected loss of $\hat{\theta}$ can be approximated by drawing a large sample size $N$ from the posterior distribution, applying a loss function $L$, and averaging over the number of samples (Davidson-Pilon, 2015):

$$\frac{1}{N} \sum_{i=1}^{N} L(\theta_i, \hat{\theta}) \approx E_\theta[L(\theta, \hat{\theta})] = l(\hat{\theta}). \tag{3}$$

Hereby, we can approximate the expected loss $l$ for every possible estimate $\hat{\theta}$ (every decision we can make) according to the loss function in use. Minimization of a loss function returns a point estimate known as a Bayes action or a Bayesian estimator, which is the decision with the least expected loss according to the loss function and the decision in which we are interested in this work (Berger, 2013; Moyé, 2006).

### 2.1.2 Customization of our case-specific loss function

Davidson-Pilon (2015) and Hennig and Kutlukaya (2007) have proposed that it might be useful to move on from standard objective loss functions to the design of customized loss functions that specifically reflect an individual's (i.e., the decision maker's) objectives and preferences regarding outcomes. Especially as we assign an economic notion to geological models and related estimation problems, we argue that it is necessary to consider the subjective perspectives of involved decision makers, for example exploration and production companies. Consequently, the design of a more specific nonstandard and possibly asymmetric loss function might be required, one that includes subjective aspects and differences in weighting of particular risks, arising from a decision maker's inherent preferences and the environment in which this actor has to make a decision. In the face of several uncertain parameters, which is a given in complex geological models, a perfect estimate, a perfect decision, is virtually unattainable. However, an attempt can be made to design a custom loss function that returns a Bayesian estimator involving the least bad consequences for a decision maker in a specific environment (Davidson-Pilon, 2015; Hennig and Kutlukaya, 2007).

Hennig and Kutlukaya (2007) argue that choosing and designing a loss function involves the translation of informal aims and interests into mathematical terms. This process naturally implies the integration of subjective elements. According to them, this is not necessarily unfavorable or less objective, as it may better reflect an expert's perspective on the situation.

Standard symmetric loss functions can easily be adapted to be asymmetric, for example by weighting errors to the negative side stronger than those to the positive side. Preference over estimates larger than the true value, i.e., overestimation, is thus incorporated in an uncomplicated way. Much more complicated designs of loss functions are possible, depending on purpose, objective, and application. We will describe potential design options in the following.

For our example of estimating the economic value of a hydrocarbon prospect, which is represented by the maximum trap volume $V_t$, we develop a custom loss function in five steps. Ideally, a decision maker would like to know the exact true value so that resources can be allocated appropriately in order to acquire economic gains by developing a project and producing from a reservoir. This conscious and irrevocable allocation of resources is the decision to be made or action to be taken (Bratvold and Begg, 2010). Thus, we treat estimating as equivalent to making a decision. Deviations from the

unknown true value in the form of over- and underestimation bring about an error and loss accordingly.

It can be assumed that several decision makers in one such environment or sector may have the same general loss function but different affinities concerning risks. This might be based, for example, on different psychological factors or economic philosophies followed by companies. It might also be based on the budgets and options such actors have available. An intuitive example is the comparison of a small and a large company. A false estimate and wrong decision might have a significantly stronger impact on a company that has a generally lower market share and few projects than on a larger company that might possess higher financial flexibility and for which one project is only one of many development options in a wide portfolio.

In steps I–IV we make assumptions about the significance of such deviations and how they differently contribute to expected losses in the general decision-making environment and introduce the concept of varying risk affinities in the final step V.

- *Step I – Choosing a standard loss function as a starting point.* In our case, we assume that investments increase linearly with linear growth in the value of the prospect. For this reason, we choose the symmetric absolute-error loss function as a basis for further customization steps:

$$L(\theta, \hat{\theta}) = |\theta - \hat{\theta}|. \tag{4}$$

- *Step II – Simple overestimation.* Considering the development of a hydrocarbon reservoir, it can be assumed that over-investing is worse than under-investing. Overestimating the size of an accumulation might, for example, lead to the installation of equipment or facilities that are actually redundant or unnecessary. This would come with additional unrecoverable expenditures. Consequences from underestimating $(0 < \hat{\theta} < \theta)$, however, may presumably be easier to resolve. Additional equipment can often be installed later on. Hence, simple overestimation $(0 < \theta < \hat{\theta})$ is weighted stronger in this loss function by multiplying the error with an overestimation factor $a$:

$$L(\theta, \hat{\theta}) = |(\theta - \hat{\theta})| \ a. \tag{5}$$

- *Step III – Critical overestimation.* The worst case for any project would be that its development is set into motion expecting a gain only to discover later that the value in the reservoir does not cover the costs of realizing the project, resulting in an overall loss. A petroleum system might also turn out to be a complete failure containing no value ($V_t = 0$ in our 3-D case study) at all, although the actor's estimate indicated the opposite. Here, we refer to this as critical overestimation. A positive value is estimated, but the true value is zero or negative

$(\theta \leq 0 < \hat{\theta})$. This is worse than simple overestimation, whereby both values are positive and a net gain is still achieved, which is only smaller then the best possible gain of expecting the true value. Critical overestimation is included in the loss function by using another weighting factor $b$ that replaces $a$:

$$L(\theta, \hat{\theta}) = |(\theta - \hat{\theta})| \ b. \tag{6}$$

In other words, with $b = 2$, critical overestimation is twice as bad as simple overestimation.

- *Step IV – Critical underestimation.* We also derive critical underestimation from the idea of estimating zero (or a negative value) when the true value is actually positive $(\hat{\theta} \leq 0 < \theta)$. This is assumed to be worse than simple overestimation but clearly better than critical overestimation. No already owned resources are wasted, and it is only the potential value that is lost, i.e., opportunity costs that arise from completely discarding a profitable project. Critical underestimation is weighted using a third factor $c$:

$$L(\theta, \hat{\theta}) = |(\theta - \hat{\theta})| \ c. \tag{7}$$

- *Step V – Including different risk affinities.* We now further adapt the loss function to consider varying risk affinities of different actors. We follow the approach of Davidson-Pilon (2015), who implemented different risk affinities by simply introducing a variable risk factor. Using different values for this factor, we can represent how comfortable an individual is with being wrong and furthermore which "side of wrong" is preferred by that decision maker (Davidson-Pilon, 2015). In our case, bidding lower is considered the cautious, risk-averse option, as smaller losses can be expected from underestimating. Guessing higher is deemed riskier, as losses from overestimation are greater. However, bidding correctly on a higher value will also return a greater gain. It is assumed that risk-friendly actors care less about critical underestimation; i.e., they would rather develop a project than discard it. In our finalized loss function, we simply include these considerations via a risk affinity factor $r$, which alters the incurred losses:

$$L(\theta, \hat{\theta}) = \begin{cases} |\theta - \hat{\theta}| \ r^{-0.5}, & \text{for } 0 < \hat{\theta} < \theta \\ |\theta - \hat{\theta}| \ a \ r, & \text{for } 0 < \theta < \hat{\theta} \\ |\theta - \hat{\theta}| \ b \ r, & \text{for } \theta \leq 0 < \hat{\theta} \\ |\theta - \hat{\theta}| \ c \ r^{-0.5}, & \text{for } \hat{\theta} \leq 0 < \theta \end{cases},$$

with $a, b, c, r \in \mathbb{Q}$. $\tag{8}$

This equation shows that the final custom loss function is in essence a composite of four different functions for the over- and underestimation cases explained in steps II–IV. It is important to note that the weighting factors $a$, $b$, and $c$ can take

basically any numerical values but should be chosen in a way that they appropriately represent the framework conditions of the problem. Here, we assume that simple overestimation is 25 % ($a = 1.25$), critical overestimation 100 % ($b = 2$), and
5 critical underestimation 50 % ($c = 1.5$) worse than simple underestimation.

According to Eq. (8), the risk-neutral loss function is returned for $r = 1$, as no reweighting takes place. For $r < 1$, the weight on overestimating ($a$, $b$) is reduced and increased
for critical underestimation ($c$), as well as normal underestimation. This represents a risk-friendlier actor that is willing to bid on a higher estimate to attain a greater gain. For $r > 1$, the overestimation weight ($a$, $b$) is increased in the loss function, underestimation and critical underestimation weight ($c$)
are decreased, and more risk-averse actors are prompted to bid on lower estimates. Since risk neutrality is expressed by $r = 1$, we consider values $0 < r < 2$ to be the most appropriate choices to represent both sides of risk affinity equally.

In Fig. 1, we illustrate different aspects and steps of adapt-
20 ing and applying the custom loss function. For these simple examples, we assume that the economic value of our reservoir is represented by an abstract score parameter. Figure 1a depicts the plotting of the absolute-error loss function (customization step I) applied to a normal distribution. It can be
seen that for this standard symmetrical function, the minimal point of expected losses and Bayes action corresponds to the median (and mean for this symmetric distribution). Figure 1b summarizes customization steps II–IV and visualizes how four different functions for four cases of under- and
overestimation are summed up to one combined loss function that comprises all of the assumptions made for the decision-making environment. A jump of expected losses on the negative side of possible estimates can be attributed to the way we defined the function for critical underestimation as dependent
on zero.

In Fig. 1c, the risk factor of step V was implemented without steps II–IV, i.e., only for the standard absolute-error loss function. It can be seen that risk-averse and risk-friendly decision makers are represented by different realizations of ex-
40 pected losses based on one and the same normal distribution: the narrow shape of the risk-friendly function represents improved confidence in the decision, while the increased expected loss (Bayes risk) of the minimum indicates that this comes along with the acceptance of a higher risk. Inversely,
the flat shape of the risk-averse function can be seen as reduced confidence in the decision. There is less of a difference in making a different decision than for the risk-friendly actor. At the same time, the expected loss of the minimum, and thus the accepted risk, is lower. However, although they differ
in expected losses, both decision makers share the same individual best estimate, since the loss function in itself is still symmetric. This changes in panel (d), in which all customization steps were applied. Here, the risk factor reweights the influence of the subfunctions shown in panel (b). Under- and
overestimation cases are accordingly enhanced or reduced in

impact so that the resulting loss function becomes asymmetric and minima are found at different score estimates, given the same underlying information.

In Fig. 1e and f, the functions from panels (c) and (d) are applied on a score distribution resulting from the combina-
60 tion of two other uncertain parameters: reservoir thickness and depth. This can be seen as an extremely simplified 1-D model with only two inputs that define one output as a parameter of interest, the final score. In this case, thickness is seen as the potential positive value in our reservoir, as it provides
space for hydrocarbons to accumulate. Depth is subtracted from this, as it implies a cost of drilling. Thus, the final score is a very essential representation of the economic value given the information available. The respective final distribution is slightly skewed. Figure 1e depicts the respective application
of the same functions used in panel (c): symmetric, but including risk affinity. The overall effects are the same as in panel (c). It can be additionally observed that since the underlying distribution is now asymmetric, all expected loss minima are found on the median estimate, lower than the mean.
In panel (f), the complete custom loss function was applied as in panel (d). Based on the uncertain information about the final score, the three differently risk-affine loss functions plot differently, with minima in the negative space, at zero, and in the positive space. This illustrates how the risk-averse de-
cision maker tends to expect a possible negative outcome, while the risk-friendly actor bids on a positive value. This could be seen as the decision to abandon versus the decision to invest in a prospect.

For a better understanding of how our finalized custom
loss function determines the incurrence of loss, actual losses for three fixed true values and risk neutrality ($r = 1$) are plotted in Fig. 2.

It has to be emphasized that this is just one possible proposal for loss function customization. There is not one per-
90 fect design for such a case (Hennig and Kutlukaya, 2007). Slight to strong changes can already be implemented by simply varying the values of the weighting factors $a$, $b$, and $c$. Fundamentally different loss functions can also be based on a significantly different mathematical structure. As loss func-
95 tions are customized regarding the problem environment and according to the subjective needs and objectives of the decision maker, they are mostly defined by the actor expressing his or her perspective (Davidson-Pilon, 2015; Hennig and Kutlukaya, 2007). Changes in the individual's perception and
100 attitude might lead to further customization needs at a future point in time, as reported by Hennig and Kutlukaya (2007).

## 2.2 Case study: synthetic 3-D structural geological model

Next we want to show that this loss function approach is not
only applicable to simple probability distributions but is an equally useful tool to estimate the true value of a parameter of interest resulting from more complex geological models that

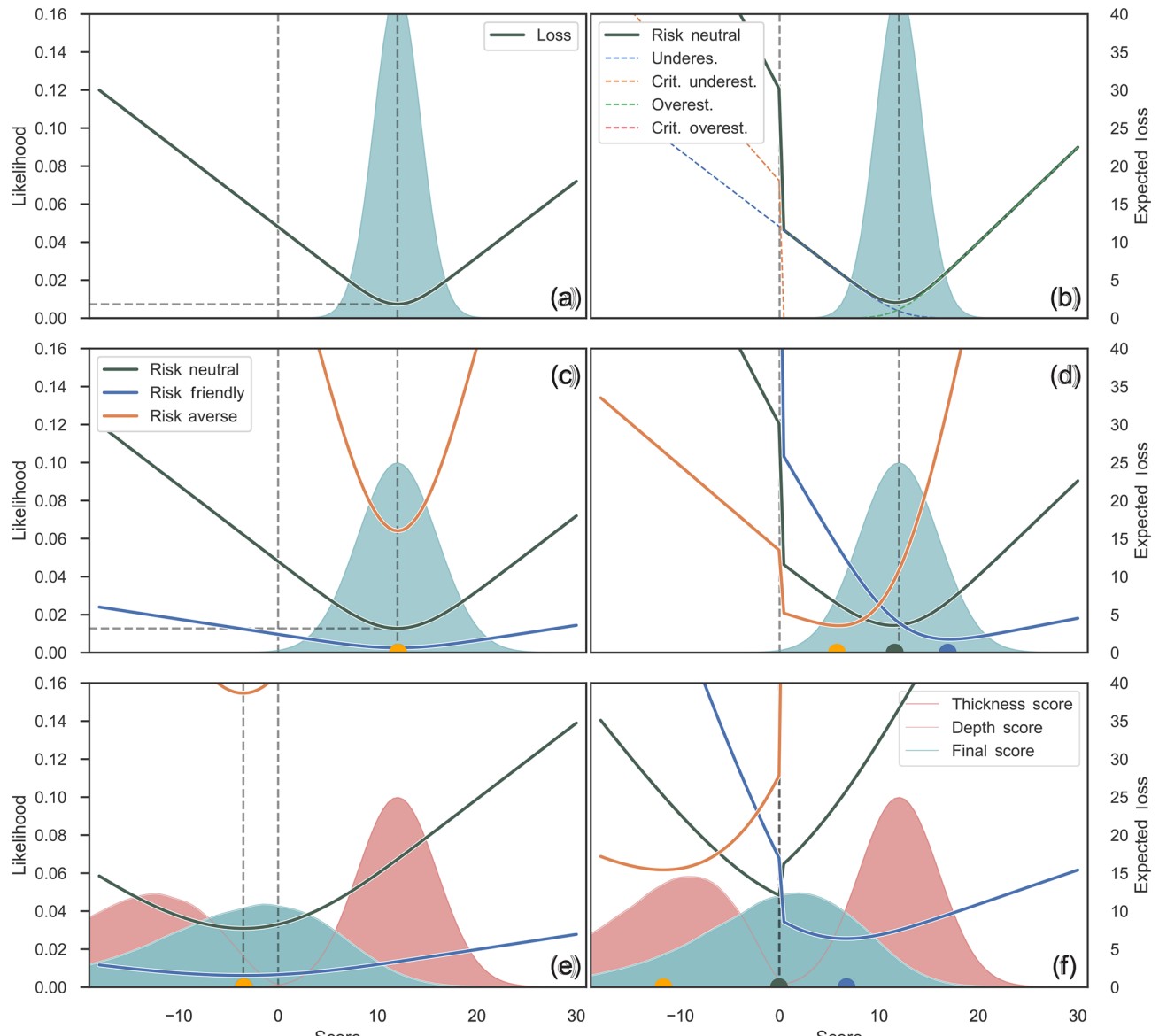

**Figure 1.** Illustration of different steps and aspects of our loss function customization. Functions are applied to an abstract score as the parameter of interest.

encompass numerous uncertain input parameters. As a case study, we now consider a synthetic 3-D structural geological model that is placed in a probabilistic framework.

### 2.2.1 Computational implementation

5 Computationally, we implement all of our methods in a Python programming environment, relying in particular on the combination of two open-source libraries: (1) GemPy (version 1.0) for implicit geological modeling and (2) PyMC (version 2.3.6) for conducting probabilistic simulations.

10 GemPy is able to generate and visualize complex 3-D structural geological models based on a potential-field interpolation method originally introduced by Lajaunie et al.

(1997) and further elaborated by Calcagno et al. (2008). GemPy was specifically developed to enable the embedding of geological modeling in probabilistic machine-learning 15 frameworks, in particular by coupling it with PyMC (de la Varga et al., 2019).

PyMC was devised for conducting Bayesian inference and prediction problems in an open-source probabilistic programming environment (Davidson-Pilon, 2015; Salvatier 20 et al., 2016). Different model-fitting techniques are provided in this library, such as various Markov chain Monte Carlo (MCMC) sampling methods. For our purpose we make use of adaptive metropolis sampling by Haario et al. (2001) and check MCMC convergence via a time series method ap- 25

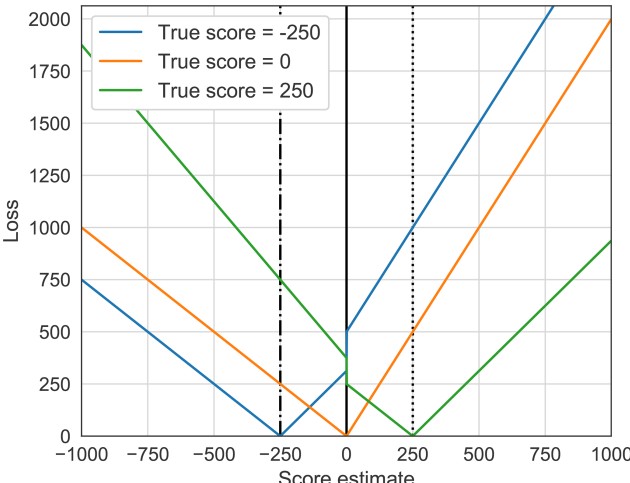

**Figure 2.** Loss based on the risk-neutral custom loss function (Eq. 8) for determined true scores of −250, 0, and 250. This plot is meant to clarify the way real losses are incurred for each estimate relative to a given true value. The expected loss, as seen in Fig. 1, is acquired by arithmetically averaging over all deterministic loss realizations based on the score probability distribution by using Eq. (3).

proach by Geweke et al. (1991). Components of a statistical model are represented by deterministic functions and stochastic variables in PyMC (Salvatier et al., 2016). We can thus use the latter to represent uncertain model input parameters and link them to additional data via likelihood functions. Other parameters, such as the value of interest for decision-making, can be determined over deterministic functions as children of parent input parameters.

To visually compare the states of geological unit probabilities after conducting stochastic simulations, we consider the normalized frequency of lithologies in every single voxel and visualize the results in probability fields (see Wellmann and Regenauer-Lieb, 2012).

### 2.2.2 Design of the 3-D structural geological model

Our geological example model is designed to represent a potential hydrocarbon trap system. Stratigraphically, it includes one main reservoir unit (sandstone), one main seal unit (shale), an underlying basement, and two overlying formations that are assumed to be permeable so that hydrocarbons could have migrated upwards. Structurally, it is constructed to feature an anticlinal fold that is displaced by a normal fault. All layers are tilted and dip in the opposite direction of the fault plane dip. A potential hydrocarbon trap is thus found in the reservoir rock enclosed by the deformed seal and the normal fault.

Using GemPy, we construct the geological model as follows: in principle, it is defined as a cubic block with an extent of 2000 m in the $x$, $y$, and $z$ directions. The basic input data for the interpolation of the geological features is composed of

**Table 1.** Input parameter uncertainties defined by distributions with respective means $\mu$, standard deviations $\sigma$, and shape factor $\alpha$.

|  | $\mu$ | $\sigma$ | $\alpha$ |
| --- | --- | --- | --- |
| Overlying | 0 | 40 | 0 |
| Sandstone 2 | 0 | 60 | 0 |
| Seal | 0 | 80 | 0 |
| Reservoir | 0 | 100 | 0 |
| Fault offset | 0 | −150 | −2 |

3-D point coordinates for layer interfaces and fault surfaces, as well as orientation measurements that indicate respective dip directions and angles. From these data, GemPy is able to interpolate surfaces and compute a voxel-based 3-D model (see Fig. 3).

We include uncertainties by assigning them to the $z$ positions of points that mark layer interfaces in the 3-D space. This is achieved via probability distributions (PyMC stochastic variables) from which error values are drawn. These are then added to the original input data $z$ value. As the $z$ position is the most sensible parameter for predominantly horizontal layers, we can hereby not only implement uncertainties regarding layer surface positions in depth, but also layer thicknesses, geometrical shapes, and degree of fault offset.

Such probability distributions can also be allocated as homogeneous sets to point and feature groups that are to share a common degree of uncertainty (see Table 1). We assign the same base uncertainty to groups of points belonging to the same layer bottom surface by referring them to one shared distribution each. Assuming an increase in uncertainty with depth, standard deviations for the shared distributions are increased for deeper formations. Furthermore, uncertainty regarding the magnitude of fault offset is incorporated by adding a skewed normal probability distribution that is shared by all layer interface points in the hanging wall. A left-skewed normal distribution is chosen to reflect the nature of throw on a normal fault, in particular the slip motion of the hanging wall block. Skew to the negative side ensures that the offset nature of the normal fault is maintained and inversion to a reverse fault is avoided.

This model was designed for the primary purpose of testing our loss function method. All features, uncertainties, and parameter relations were implemented in a way that they result in model variability and complexity that is adequate and significant to the decision problem in this work. The model is not aimed at representing a completely plausible or realistic geological setting.

### 2.2.3 $V_t$ as the parameter of interest

Given full 3-D representation of geological structures, we can now define the trap volume $V_t$ as the parameter of interest, a feature that indicates the economic value of the reservoir in this case. For conducting straightforward volumetric

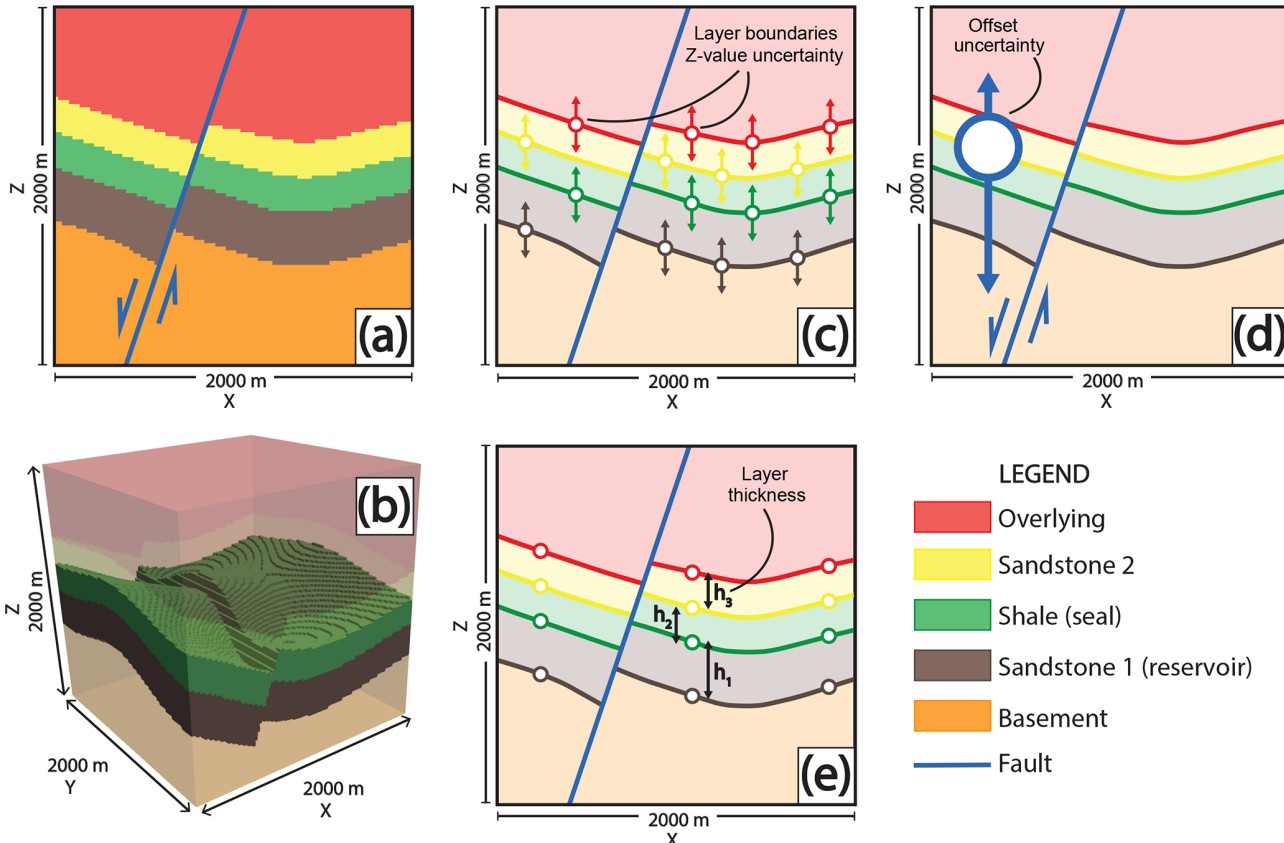

**Figure 3.** Design of the 3-D structural geological model. A 2-D cross section through the middle of the model ($y = 500$ m), perpendicular to the normal fault (parallel to the $x$–$z$ plane), is shown in **(a)**. A 3-D voxel representation of the model, highlighting the reservoir and seal formations, is visualized in **(b)**. In **(d)** and **(d)**, the inclusion of parameter uncertainties is presented. Colors indicate certain layer bottoms (i.e., boundaries) that are assigned shared $z$-positional uncertainties **(c)**. All points in the hanging wall are additionally assigned a fault offset uncertainty **(d)**. Thicknesses of the three middle layers are defined by the distances of boundary points **(e)** and are thus directly dependent on **(c)**.

calculations, we assume that closed traps are always filled to spill; i.e., we only consider structural features as controlling mechanisms and disregard other parameters in the OOIP equation (Eq. 9).

We argue that $V_t$ can be inserted for the hydrocarbon-filled rock volume $A \cdot h$ TS2 in the OOIP equation (Dean, 2007; Morton-Thompson et al., 1993):

$$\text{OOIP} = A \cdot h \cdot \phi \cdot (1 - S_W) \cdot 1/\text{FvF}, \tag{9}$$

where OOIP is returned in cubic meters, $A$ is the drainage area, $h$ the net pay thickness, $\phi$ the porosity, $S_W$ the water saturation, and FvF the formation volume factor that determines the shrinkage of the oil volume brought to the surface.

By declaring these connections, we have given our model an economic significance. We can assume that the hydrocarbon trap volume is directly linked to project development decisions; i.e., the investment and allocation of resources is represented by bidding on a volume estimate.

In the course of this work, we developed a set of algorithms to enable the automatic recognition and calculation of trap volumes in geological models computed by GemPy. The volume is determined on a voxel-counted basis via four conditions illustrated in Fig. 4 and further explained in Appendix A.

Following these conditions, we can define four major mechanisms that control the maximum trap volume: (1) the anticlinal spill point of the seal cap, (2) the cross-fault leak point at a juxtaposition of the reservoir formation with itself, (3) leakage due to juxtaposition with overlying layers and cross-fault seal breach (failure related to the shale smear factor, SSF), and (4) stratigraphical breach of the seal when its voxels are not continuously connected above the trap. Due to the nature of our model, (3) and (4) will always result in complete trap failure. The occurrence of these trap control mechanisms can be tracked throughout stochastic simulations of the model.

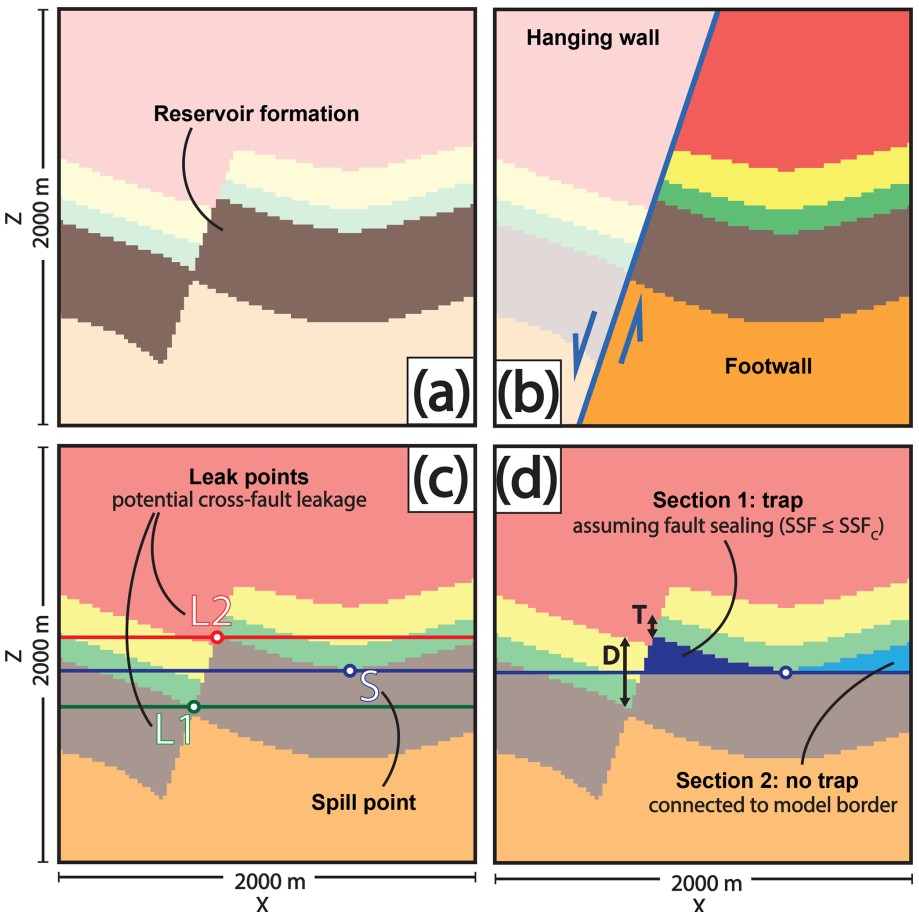

**Figure 4.** Illustration of the process of trap recognition in 2-D, i.e., the conditions that have to be met by a model voxel to be accepted as belonging to a valid trap. A voxel has to be labeled as part of the target reservoir formation **(a)** and positioned in the footwall **(b)**. Trap closure is defined by the seal shape and the normal fault **(c)**. Consequently, the maximum trap fill is defined by either the anticlinal spill point (S) or a point of leakage across the fault, depending on juxtapositions with layers underlying (L1) or overlying the seal (L2). The latter is only relevant if the critical shale smear factor is exceeded, as determined over $D$ and $T$ in **(d)**. In this example, assuming sealing of the fault due to clay smearing, the fill horizon is determined by the spill point in **(d)**. Subsequently, only trap section 1 is isolated from the model borders in **(d)** and can thus be considered a closed trap. Voxels included in this section are counted to calculate the maximum trap volume.

### 2.2.4 Generating different probability distributions for $V_t$

The trap volume $V_t$ is a result from GemPy's implicit geological model computation. It is an output parameter dependent on deterministic and stochastic input parameters. When conducting stochastic simulations, input uncertainties will propagate to $V_t$, which is thereby represented by a respective probability distribution that our custom loss function can be applied to. Using simple Monte Carlo error propagation, with every iteration, we draw sample values for our uncertain primary model input parameters defined in Sect. 2.2.2, and thus, with every iteration, we create one possible realization of our geological model, which in turn comes with one possible outcome for $V_t$. Results from all iterations together approximate the probability distribution for $V_t$ according to the input parameters.

Furthermore, we consider the possibility of updating our model by adding additional secondary information via Bayesian inference. We do this by introducing likelihood functions that constrain our primary parameters. We have to note that these inputs remain unchanged; however, their prior probability distributions are revalued given the additional statistical information. We achieve this by conducting Markov chain Monte Carlo (MCMC) simulations. Decision-making is then based on the resulting posterior probability. Using different likelihood functions, we can create and generate different posterior probability distributions for $V_t$, which represent different information scenarios. Since we use Bayesian inference to revalue our original prior inputs, we can compare all outcomes and realizations of our custom loss function.

For the application of Bayesian inference, we implement two types of likelihoods.

1. *Layer thickness likelihoods.* With every model realization, we extract the $z$ distance between layer boundary input points at a central $x-y$ position ($x = 1100$ m, $y = 1000$ m) in our input interpolation data. Resulting thicknesses can then be passed on to stochastic functions in which we define thickness likelihoods via normal distributions.

2. *Shale smear factor (SSF) likelihood.* SSF values are realized over more complex parameter compositions. We base this likelihood on a normal distribution that we link to the geological model output.

The inclusion of these likelihoods is based on purely hypothetical assumptions and is intended to provide the opportunity to explore the effects that different types and scenarios of additional information might have. While the thickness likelihood functions are dependent on input parameters directly, the implementation of the SSF likelihood function requires a full computation of the model and extended algorithms of structural analysis.

Although Bayesian inference was utilized in this case study, it served primarily for the generation of these different but comparable distributions on which to base our decision-making, i.e., the application of our custom loss function. For additional information on how implicit geological modeling can be embedded in a Bayesian framework and how this can be used to reduce uncertainty, we refer to the work by Wellmann et al. (2010b), de la Varga and Wellmann (2016), de la Varga et al. (2019), and Wellmann et al. (2017).

## 3 Results

We applied our custom loss function to various different $V_t$ probability distributions resulting from stochastic simulations. First, reference results were created using only primary inputs (priors) and simple Monte Carlo error propagation (10 000 sampling iterations, Scenario 1). Then we devised several scenarios of additional information and included these via likelihoods and Bayesian inference. For this, 10 000 MCMC sampling steps were conducted, with an additional burn-in phase of 1000 iterations. The prior parameter uncertainties were chosen to be identical for all simulations (see Table 1). Results of convergence diagnostics can be found in Appendix C.

We present the following information scenarios.

1. *Prior-only* model

2. Introducing *seal thickness likelihoods*

    a. Likely thick seal

    b. Likely thin seal

3. Introducing *reservoir thickness likelihoods*

    a. Likely thick reservoir

    b. Likely thick reservoir and thick seal

4. Introducing *SSF likelihoods*

    a. SSF likely near its critical value

    b. Likely reliable SSF and thick seal

The implemented likelihoods are listed in Table 2.

For the comparison of results, we consider in particular the following measures: (1) probability field visualization, (2) occurrence of trap control mechanisms, (3) resulting trap volume distributions, and (4) consequent realization of expected losses and related decisions.

### 3.1 Prior-only model (Scenario 1)

Probability field visualization illustrates well how the prior uncertainty is based on normal distributions (see Fig. B2). Trap control mechanisms are listed in Table B2. For this prior-only scenario, all four relevant mechanisms occur. The dominant factor is the anticlinal spill point with a 51.5 % rate of occurrence. It is followed by cross-fault leakage to the reservoir (25 %) and other permeable formations (12 %). Stratigraphical breaches of the seal were registered to be decisive in about 11 % of iterations. In only 0.5 % of iterations, the algorithm failed to recognize a mechanism; i.e., correct model realization failed.

Maximum trap volumes were calculated for each model iteration and plotted as a probability distribution in Fig. 5. In general, a wide range of volumes is possible, from zero to more than 3 million m$^3$. However, we can recognize a bimodal tendency: low volumes are less probable than significantly high volumes or complete failure ($V_t = 0$).

Consequently, applying our custom loss function to this distribution resulted in widely separated minimizing estimators for the differently risk-inclined actors (see Fig. 5). Only the risk-friendliest estimates are found within the described highly positive mode of the distribution. Risk-averse individuals bid on significantly lower estimates or even zero. The risk-neutral decision is found between the two modes and presents the highest expected loss. Expected losses decrease towards the extreme decisions and closer to the modes.

### 3.2 Introducing seal thickness likelihoods (Scenarios 2a and 2b)

We considered two scenarios of thickness likelihoods: the seal being (Scenario 2a) likely very thick or (Scenario 2b) likely very thin (see Table 2).

In Scenario 2a, probability visualization illustrates that the presence of a thick seal is very probable (see Fig. B2). For Scenario 2b, the presence of a reliable seal is questionable.

A high likelihood of a reliable seal cap (2a) significantly reduced the probability of trap failure, while enhancing the mode of highly positive outcomes (see Fig. 5). This coincides with the predominance of the anticlinal spill point (63 %) and

**Table 2.** Normal distribution mean ($\mu$) and standard deviations ($\sigma$) for the likelihoods implemented in the different scenarios.

|  | Seal thickness | | Reservoir thickness | | SSF | |
|---|---|---|---|---|---|---|
|  | $\mu$ (m) | $\sigma$ (m) | $\mu$ (m) | $\sigma$ (m) | $\mu$ | $\sigma$ |
| Scenario 1 | – | – | - | – | – | - |
| Scenario 2a | 300 | 30 | - | – | – | - |
| Scenario 2b | 50 | 30 | - | – | – | - |
| Scenario 3a | 350 | 30 | - | – | – | - |
| Scenario 3b | 300 | 30 | 300 | 30 | – | - |
| Scenario 4a | – | – | - | – | 5.1 | 0.3 |
| Scenario 4b | 300 | 30 | - | – | 2 | 0.3 |

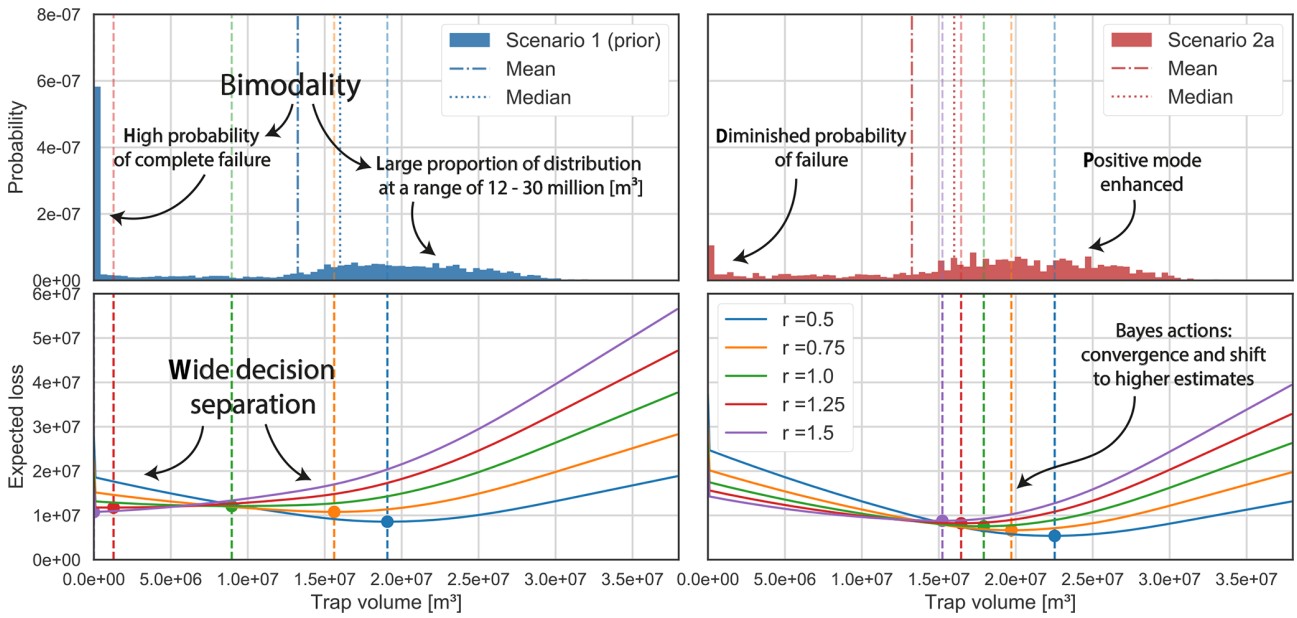

**Figure 5.** Trap volume distribution and resulting loss function realizations for Scenario 1 (prior) and Scenario 2a, in which we introduced the likelihood of a thick seal. Comparing both, we can observe how the additional information reduced the bimodality in the posterior distribution (2a), particularly by reducing the probability of complete failure and enhancing positive probabilities. Consequently, Bayes actions converged and expected losses were reduced.

the leak point to the same reservoir (36 %) as control mechanisms. The occurrence of other mechanisms was negligible (see Table B2). Inversely, a likely thin seal (2b) virtually eliminated the positive mode and focused almost the whole distribution on complete failure. Accordingly, seal-breach-related control mechanisms gained importance (65.5 % occurrence rate for stratigraphical seal breach).

In both scenarios, Bayes actions shifted towards the respectively emphasized modes. This came with the overall convergence of decisions and reduction of expected losses. In Scenario 2a, all decision makers bid on a positive outcome. Risk-averse individuals experienced the strongest shift but also present the highest expected losses. In Scenario 2b, all individuals decide not to allocate resources. Even the risk-friendliest actor moved to a zero estimate, with the most risk-averse bid having already been placed in the prior Scenario 1.

However, although all decisions coincide, expected losses increase from risk averse to risk friendly (see Table B1).

## 3.3 Introducing reservoir thickness likelihoods (Scenarios 3a and 3b)

We also tested scenarios for the likelihood of a thick reservoir formation alone (Scenario 3a) and in combination with the likelihood of a thick seal (Scenario 3b; see Table 2). The overall effect of using these reservoir-based likelihoods turned out to be minor compared to the seal-related scenarios.

In Scenario 3a, failure probabilities slightly increased, resulting in a decision shift towards lower values (see Fig. B1). Results for Scenario 3b are very similar to those of 2b, as can also be seen in Table B1. There was no significant reduction of expected losses or a shift in decisions by adding

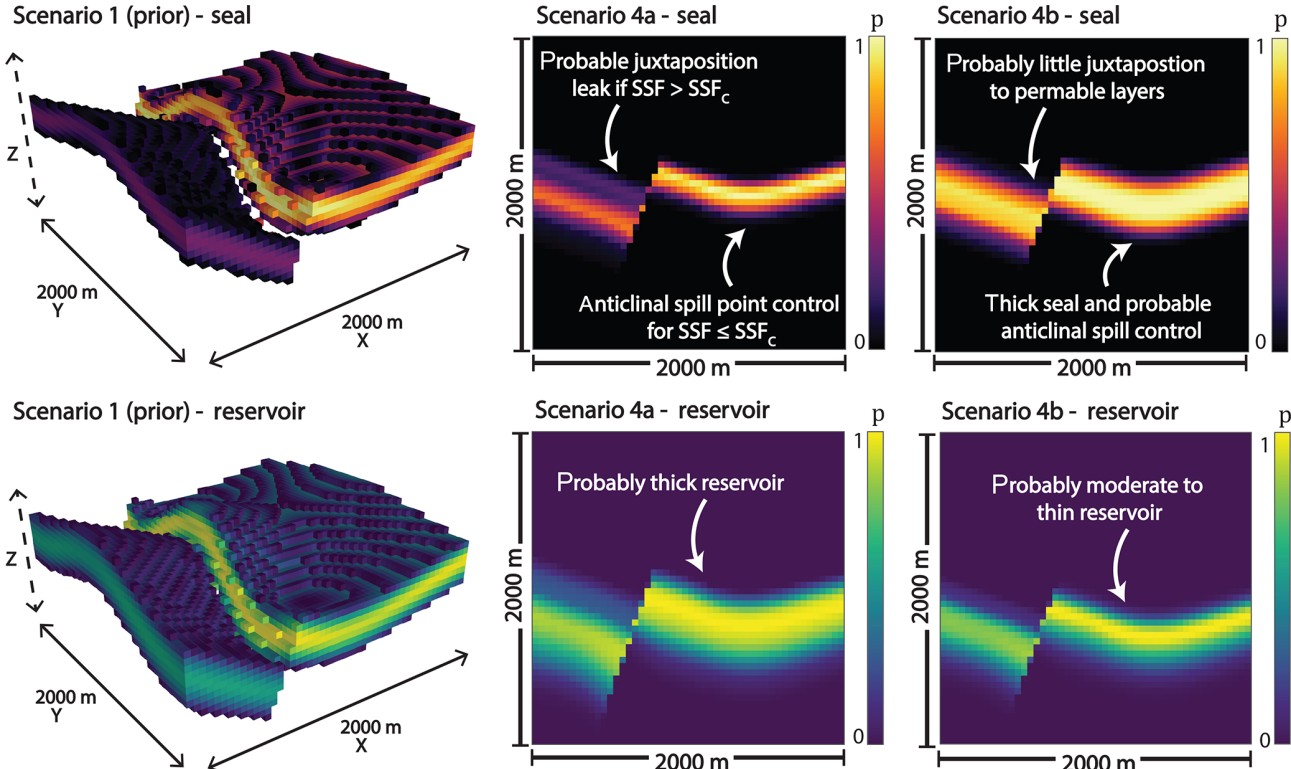

**Figure 6.** Probability field visualizations for seal and reservoir units in Scenarios 1 (prior), 4a, and 4b. For Scenario 1, we used 3-D voxel visualizations and set a threshold at a probability of 0.5 (only voxels with a probability higher than 0.5 are shown). It can be recognized that the seal is disrupted across the fault in more than 50 % of the prior model realizations. For the other scenarios, we show the full probability field for both units on a section through the middle of the model ($y = 500$ m), parallel to the $x$–$z$ plane.

the likelihood of a thick reservoir to the likelihood of a thick seal.

### 3.4   Introducing SSF likelihoods

We considered two SSF-related likelihood scenarios. In Scenario 4a, we implemented solely an SSF likelihood that was based on a narrow normal distribution ($\mu = 5.1$, $\sigma = 0.3$) with a mean near the critical value $SSF_c = 5$. In Scenario 4b, we combined the likelihood of a thick seal (2a) with a likely moderate but reliable SSF value (SSF normal distribution with $\mu = 2$ and $\sigma = 0.3$). Figure 6 illustrates the posterior situations well.

Scenario 4a resulted in increased bimodality of the posterior distribution (see Fig. 7). Accordingly, the Bayes action divergence and expected losses increased. Only two trap control mechanisms remained relevant for 4a (see Table B2): anticlinal spill (66 %) and cross-fault leakage to overlying formations (34 %).

The results for 4b were comparable to those of 2a but more pronounced. Entropies, particularly related to the seal thickness, were clearly reduced, also in the hanging wall. Probabilities of failure and low volumes were almost eliminated, further enhancing the highly positive mode. This con-

sequently resulted in an even higher convergence of Bayes actions, as well as reduction of expected losses compared to Scenario 2a. Anticlinal spill is the decisive control mechanism in 79.5 % of cases; otherwise, only cross-fault leakage to the reservoir occurred (20.5 %).

### 4   Discussion

Our results show that it is possible to apply Bayesian decision theory to geological models as an approach to obtain an objective basis for decisions by considering uncertainties in these models. Even though the concept itself is not new, the application to the context of probabilistic geological modeling requires some adaptation and care when constructing appropriate loss functions. Our results highlight the potential use of custom loss functions, first for a simplified 1-D case, and then for a more complex full 3-D model. Even though these models are both conceptual, they highlight in our point of view the interesting potential of the method, as the optimal decision, the Bayes action, is not always directly obvious when only considering posterior predictive distributions. The addition of subjective risk affinity and the risk of critical overestimation particularly lead to interesting changes in the

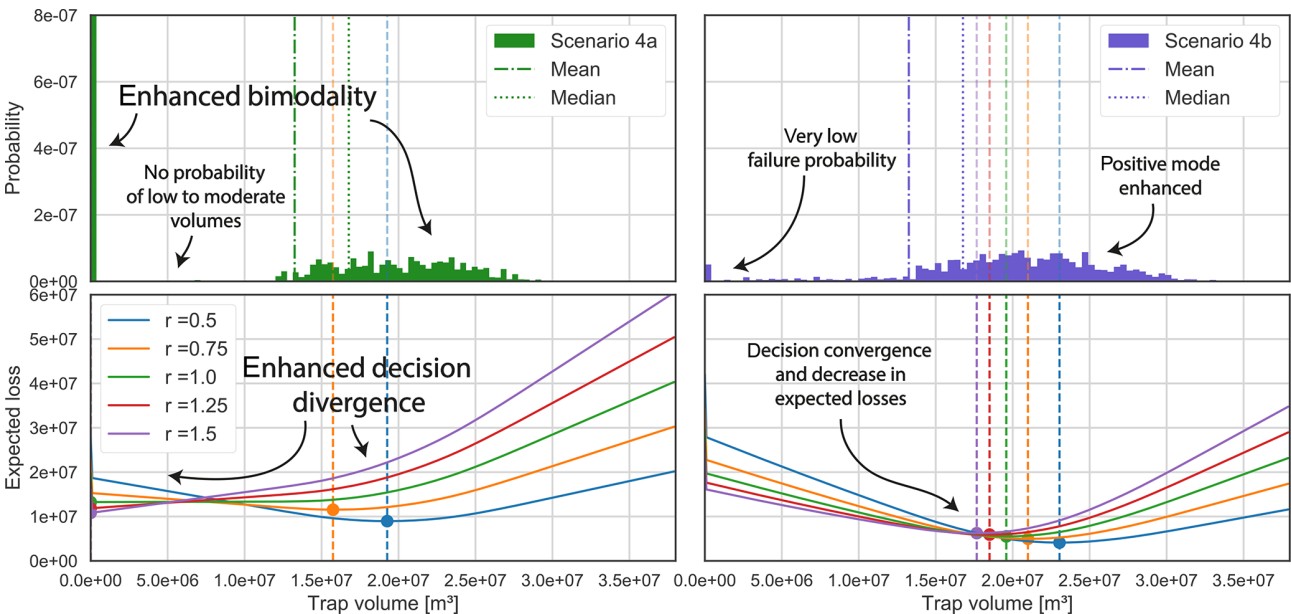

**Figure 7.** Trap volume distribution and resulting loss function realizations for Scenario 4a and Scenario 4b. Adding a likelihood of the SSF being around its critical value led to increased bimodality and an elimination of low to moderate volume probabilities. Bayes actions diverged accordingly in Scenario 4a. Implementing a reliable SSF value likelihood ($\mu = 2, \sigma = 0.3$) in combination with the thick seal likelihood from Scenario 2a resulted in a emphasis on highly positive volumes. This, in turn, led to a stark convergence of decisions and reduction of expected losses.

optimal decision. Given these aspects, we consider the use of custom loss functions with probabilistic geological modeling to be a very suitable combination in the framework of Bayesian decision theory.

The case study considered here addressed a typical scenario of exploration for a fluid reservoir. We first discuss additional relevant points with regard to this specific case and then provide more general comments on extensions and the application in additional fields in which geological models are commonly used.

### 4.1 On the impact of additional information on decision-making

We used these loss-function-related indicators to assess the significance of additional information on a decision. We observed that the impact on decision uncertainty, induced by Bayesian inference, is not simply strictly aligned with the change in uncertainty regarding model parameters but on parameter combinations that are relevant for the outcome of the value of interest. It seems to be of central importance (1) "where" in the model uncertainty is reduced, i.e., in which spatial area or regarding which model parameters, and (2) which possible outcome is enhanced in terms of probability. An increased probability of a thick or thin seal in our model equally reduced decision uncertainty significantly by raising the probability of a positive or negative outcome, respectively. Improved certainty about our reservoir thickness, however, had far lesser impact on decision-making. This

shows that some areas and parameter combinations have a much greater influence on the decision uncertainty than others, depending on the way they contribute to the outcome of the value of interest.

Some types of additional information could even lead to increased decision uncertainty. We observed this in Scenario 4a. The introduced SSF likelihood practically constrained our geological model to two possible situations: (1) a trap that is sealed off from juxtaposing layers and full to spill and (2) complete failure of the trap due to a breached seal across the fault. This made the decision problem a predominantly binary one and split the outcome distribution into two narrowed but distant modes. The resulting increase in decision divergence and expected losses show that, in some cases, adding information might leave actors in greater disagreement than before.

However, we furthermore have to consider that actors weight possible outcomes of the value distribution differently. They are consequently affected differently by the same type of information. Risk-friendly actors were the most robust in their decision-making in the face of possible trap failure. Eliminating this risk proved to be far less significant for the most risk-friendly than for risk-averse actors. Accordingly, it should be of foremost importance for risk-averse actors to reduce the uncertainty regarding critical factors, such as seal integrity, which might decide between the success and complete failure of a project. This is less relevant for risk-friendly decisions makers, who might acquire a

comparable benefit from knowing more about the probability of positive outcomes. They are less afraid of failure than they are of missing out on opportunity.

Crucial risks might be easily assessed if they are dependent on only one or a few parameters, such as seal thickness. In other cases, they are derived from more complex parameter interrelations, as is the case for the shale smear factor. To approach an effective mitigation of high risks, the complexities behind decisive factors need to be assessed thoroughly, and respective parent parameters, as well as their interdependencies, need to be identified. This might enable a better understanding of which type of information is missing and where in the model additional data might be of use for improved decision-making.

More of simply any type of information does not necessarily lead to better decisions. Instead, improved decision-making is achieved by attaining the right kind of information that is able to shed light on uncertainties that are relevant to an individual's own goals and preferences, as well as the general problem at hand. Bratvold and Begg (2010) stated that value is not generated by uncertainty quantification or reduction in itself but is created to the extent that these processes have the potential to change a decision. Such decision changes were clearly indicated by the shifting of actions in our different scenarios. According to Hammitt and Shlyakhter (1999), the difference in expected payoff between the prior and posterior optimal decision gives the expected value of information. This raises the question of to what extent a change in expected losses in itself might be an indicator for the value of information and if there is value in gaining confidence in a decision, even though it remains unchanged.

While Monte Carlo simulation is by now common in the hydrocarbon sector, it does not make decisions, as Murtha et al. (1997) emphasized – it merely prepares for it. We believe that loss functions have the potential to go one step further. A hypothetical ideal loss function would consider all conditions in an economic environment, as well as perfectly represent the preferences and goals of an actor and consequently be able to automatically find an optimal decision. While this is obviously unrealistic, we presume that an elaborate loss function might at least provide a very good preliminary decision recommendation. It might furthermore be able to weight risks that are not immediately apparent to an individual as a person. Furthermore, the influence of human biases and psychological behavioral challenges, as described by Bratvold and Begg (2010), could be mitigated.

Bayesian inference and MCMC methods have been applied for OOIP estimation and forecasting of reservoir productivity by Wadsley et al. (2005), Ma et al. (2006), and Liu et al. (2010). However, their research focused on history-matching simulations for already producing fields. Our approach of applying Bayesian inference for structural geological modeling and volumetric reservoir calculations is intended to support decision-making in the earliest stages of a reservoir when it has to be decided whether a project should

be developed or not. Nevertheless, it was shown in the research conducted by Wadsley et al. (2005) that early volumetric OOIP estimates can be combined with later calculations from production data via MCMC methods.

Our continuous approach could be integrated into common discrete decision-making frameworks, such as decision trees. In real cases, normally only a limited number of options is given. In the context of hydrocarbon exploration and production, this would relate to fixed magnitudes of resource allocation, such as a certain number of required drilling wells or the size of a production platform. Based on such previously defined actual options, we could discretize our value probability distribution into sections, which represent each decision scenario accordingly. Our minimizing estimators would then indicate the best discrete option for a decision maker.

## 4.2 State of knowledge, decision uncertainty, and consistent decision-making

~~We applied the concept here to a synthetic geological model. Nevertheless, it was designed to include some typical structural characteristics related to fluid reservoir systems. We developed algorithms aimed to consider the most common conditions that define structural traps. However, these conditions needed to be simplified and were implemented on a very conceptual level. Furthermore, uncertainties employed in the 3-D model related to $z$-positional values only and were thus of a primarily one-dimensional nature. It follows that no effective uncertainty concerning the overall structural shape was implemented, particularly regarding anticlinal features and the lateral position of the spill point.~~

As we defined trap volume to be in essence a deterministic function of uncertain model input parameters, uncertainties propagate to this parameter of interest when conducting stochastic simulations. We consider the resulting volume probability distributions to be expressions of the respective state of knowledge (or information) on which the decision-making is to be based. As this should include all parameters and conditions relevant for decision-making, we furthermore propose that the overall uncertainty inherent in this probability distribution can be referred to as "decision uncertainty" and that this entity should be viewed separately from geological model uncertainty.

By viewing decision-making as a problem of optimizing a case-specific custom loss function applied to such a state of knowledge and decision uncertainty, we were able to observe clear differences in the respective behavior of distinctly risk-inclined actors.

The position and separation of their minimizing estimators, i.e., their decisions, manifested according to the properties of the value distributions. The general spread and the occurrence of modes relative to the overall distribution and the relevant decision space appear to be particularly significant. High spread and bimodal tendencies, i.e., high overall uncertainty, resulted in a wider separation of different actions. Re-

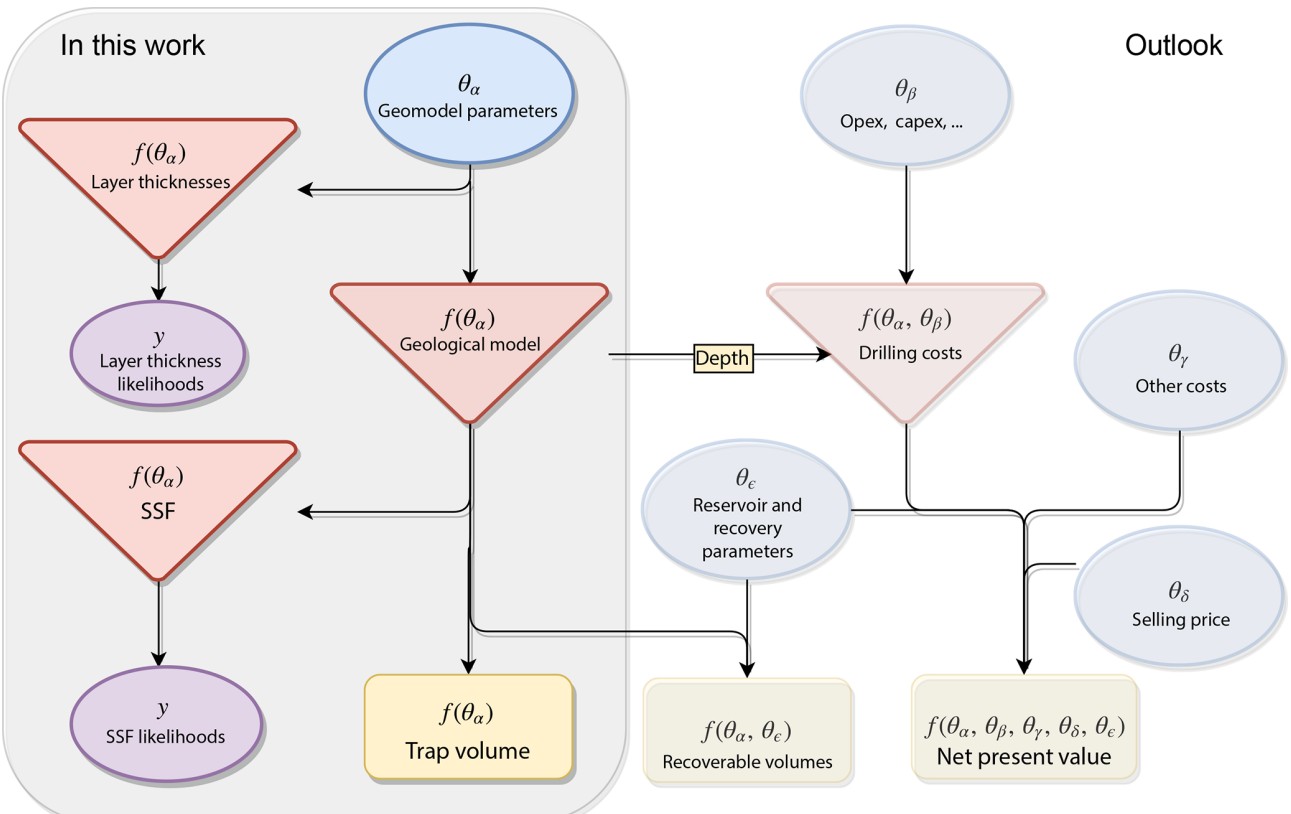

**Figure 8.** In this work, we applied our loss function approach to estimate a hydrocarbon trap volume. For this, we considered stochastic geomodeling parameters, defined deterministic functions to acquire volume, layer thicknesses, and SSF values, and linked the latter two to respective likelihoods. Regarding the bigger picture, this methodology is expandable and could include other parameters and dependencies. By taking into account other reservoir parameters and recovery factors, we could, for example, base decision-making on recoverable volumes. We could also take depth information from our model and combine this with other cost parameters to calculate drilling costs. Including additional costs, but also the selling price of hydrocarbons, we could attain the NPV as our final value of interest.

duction of the distribution to one mode conversely led to their convergence. A decrease in decision uncertainty was furthermore accompanied by a reduction in expected loss for each Bayes estimator.

5 Considering these observations, we derive the degree of action convergence and respective expected losses as measures for the state of knowledge and decision uncertainty at the moment of making a decision. The better these are, the more similar the decisions of differently risk-inclined actors 10 and the lower their loss expectations are. Given perfect information all actors would bid on the same estimate (the true value) and expect no loss, since no risk would be present. It furthermore follows from this that the relevance of risk affinity decreases with greater reduction of decision uncertainty.

15 **4.3  Extensions and outlook**

We applied the concept of decision theory here to an implicit geological modeling method (de la Varga et al., 2019). Depending on the application, other types of geometric interpolations may be more suitable to represent the geological

setting. More details on these methods, as well as the consideration of respective model uncertainties and the potential integration into probabilistic frameworks, are described, for example, in Wellmann and Caumon (2018).

We defined risk affinity to be dependent on arbitrarily chosen risk factors that led to according reweighting. Davidson-Pilon (2015) used risk parameters determined by the maximal loss each actor could incur. Other approaches could be based on more tangible values, for example by making risk attitude dependent on a fixed budget.

There are still many points that could be expanded on in future research. It would be of interest to apply the same overall concept and methodology to an authentic case based on real datasets. Given a realistic economic scenario including the capital and operational expenditures of a project, a full net-present-value (NPV) analysis could possibly be conducted by applying a loss function to an NPV distribution (see Fig. 8). A more elaborate loss function could be customized on the basis of surveys, thereby acquiring the specific preferences of one or several companies and thus ob-

taining a better profile of the economic environment, as well as the individuals acting in it.

We chose hydrocarbon systems and petroleum exploration as a sector for an exemplary application, as studies on risk related to geological modeling are most prominent in this field. However, geological modeling is of central importance to decision-making in several other fields. Directly related are all other types of subsurface fluid reservoirs, for example in groundwater extraction or geothermal energy usage. Also closely related are applications of fluid storage in subsurface reservoirs, most prominently carbon capture and storage (CCS) applications. Questions regarding storage capacity and safety deal with similar conditions and geological problems as the ones presented in this work. The described concepts can similarly be applied to other types of geological features, for example ore bodies in mineral exploration or subsurface structures and materials in geotechnical applications. In all of these cases, the geological model can have significant uncertainties and, similar to the example described in this paper, further engineering and usage aspects carry high costs. We are therefore confident that a more detailed analysis of uncertainties and the definition and understanding of custom loss functions in the context of Bayesian decision theory are very interesting paths for more research with wide possible applications.

*Code and data availability.* The code and model data used in this study are available in a GitHub repository found at http://github.com/cgre-aachen/loss_function_decision_making_paper (https://doi.org/10.5281/zenodo.2595357; Stamm, 2019 TS3 ).

## Appendix A: Determination of the maximum trap volume

The volume is calculated on a voxel-count basis. To assign model voxels to the trap feature, it is necessary to check whether the following conditions (illustrated in Fig. 4) are satisfied by each individual voxel.

1. *Labeled as reservoir formation.* The voxel has been assigned to the target reservoir formation (see Sandstone 1 in Fig. 4 (1)) in GemPy's lithology block model.

2. *Location above spill point horizon.* The voxel is located vertically above the final spill point of the trap. In the algorithm to find this final spill point, a spill point defined by the folding structure, referred to as an anticlinal spill point, and a cross-fault leak point that depends on the magnitude of displacement and the resulting nature of juxtapositions are distinguished. Once both of these points have been determined, the higher one is defined to be the final spill point used to determine the maximum fill capacity of the trap. Given a juxtaposition with layers overlying the seal, due to fault displacement, the respective section is checked for fault sealing by taking into account the shale smear factor (SSF) value, which is the ratio of fault throw magnitude $D$ to displaced shale thickness $T$ (Lindsay et al., 1993; Yielding et al., 1997; Yielding, 2012):

$$\text{SSF} = \frac{D}{T}. \tag{A1}$$

We attain both $D$ and $T$ by examining the contact between the seal lithology voxels and the fault surface.

For our model, we define the critical SSF to be $\text{SSF}_c = 5$. We assume that cross-fault sealing is breached when this threshold is surpassed. For simplicity, the fault is considered to be sealing along its plane.

3. *Location inside a closed system.* The voxel is part of a model section inside the main anticlinal feature. All of the voxels inside this particular section are separated from the borders of the model by voxels that do not meet the first two conditions above, which primarily means that they are encapsulated by seal voxels upwards and laterally. This condition is relevant under the assumption that connection to the borders of the model leads to leakage. A trap is thus defined as a closed system in this model and trap closure is assumed to be void outside the space of information, i.e., the model space. In our example model, this also means that hydrocarbons escape in the hanging wall due to respective layer dipping upwards towards the model borders.

It has to be emphasized that these conditions have been fitted to our synthetic example model. For other models featuring different geological properties, structures, and levels of complexities, these conditions and respective algorithms might not apply. Models of higher complexities will surely require the introduction of further conditions.

### A1 Anticlinal spill point detection

Regarding anticlinal structures and traps, it can be observed that, geometrically and mathematically, a spill point is a saddle point of the reservoir top surface in 3-D. This was described by Collignon et al. (2015), who pointed out that the linkage of folds is given by saddle points. These are thus a controlling factor for spill-related migration from respective structural traps. For anticlinal traps, closure can consequently be defined as the distance between the saddle point (i.e., spill point) and maximal point of the trap (Collignon et al., 2015).

Regarding a surface defined by $f(x, y)$, a local maximum at $(x_0, y_0, z_0)$ would resemble a hilltop (Guichard et al., 2013). Local maxima will be found looking at the cross sections in the planes $y = y_0$ and $x = x_0$. Furthermore, the respective partial derivatives (i.e., gradients) $\frac{\delta z}{\delta x}$ and $\frac{\delta z}{\delta y}$ will equal zero at $x_0$ and $y_0$, i.e., the extremum is a stationary point (Guichard et al., 2013; Weisstein, 2017). In the context of a geological reservoir system, such a hill can be regarded as a representation of an anticlinal structural trap. Local minima are defined analogously, presenting local minima in both planes at a stationary point (Guichard et al., 2013). A saddle point, however, is a stationary point, while not being an extremum (Weisstein, 2017). In general, saddle points can be distinguished from extrema by applying the second derivative test (Guichard et al., 2013; Weisstein, 2017): considering a 2-D function $f(x, y)$ with continuous partial derivatives at a point $(x_0, y_0)$ so that $f_x(x_0, y_0) = 0$ and $f_x(x_0, y_0) = 0$, the following discriminant $D$ can be introduced:

$$D(x_0, y_0) = f_{xx}(x_0, y_0) f_{yy}(x_0, y_0) - f_{xy}(x_0, y_0)^2. \tag{A2}$$

Using this, the following holds for a point $(x_0, y_0)$.

1. If $D > 0$ and $f_{xx}(x_0, y_0) < 0$, there is a local maximum.

2. If $D > 0$ and $f_{xx}(x_0, y_0) > 0$, there is a local minimum.

3. If $D < 0$, there is a saddle point at the point $(x_0, y_0)$.

4. If $D = 0$, the test fails (Guichard et al., 2013).

According to Verschelde (2017), a saddle point in a matrix is maximal in its row and minimal in its column. This corresponds to the logical geometrical deduction that a saddle point for a surface defined by $f(x, y)$ is marked by a local maximum in one plane but a local minimum in the perpendicular plane. In our spill point detection algorithm, we make use of GemPy's ability to return layer boundary surfaces (simplices and vertices) as well as the gradients of the potential fields in discretized arrays.

1. We first look for vertices at which the surface of interest coincides with a gradient zero point.

2. Then, we check for the change in gradient sign at each such point in perpendicular directions. If they are opposite to one another, we can classify the vertex as a saddle point.

3. Lastly, we declare the highest saddle point to be our anticlinal spill point.

## A2  Cross-fault leak point detection

For the potential point of leakage to formations underlying the seal across the normal fault (including the reservoir itself), we take the highest $z$ position of the reservoir units' contact (voxelized) with the fault in the hanging wall.

In the case of a juxtaposition with seal-overlying formations and a failed SSF check, the maximum contact of the trap with the fault becomes the final spill point. Due to the shape of the trap in our model, we can then expect full leakage and set the maximum trap volume to zero.

## A3  Calculating the maximum trap volume

When all trap voxels have been determined via the conditions defined in Sect. 2.2.3, the maximum trap volume $V_t$ is calculated by simply counting the number of trap voxels and rescaling their cumulative volume depending on the resolution in which the model was computed:

$$V_t = n_v \cdot \left( \frac{S_o}{R_m} \right)^3, \tag{A3}$$

where $n_v$ is the number of trap voxels, $S_o$ gives the original scale, and $R_m$ is the resolution used for the model.

For the example of a cubic geological model with an original extent of 2000 m in three directions, computed using a resolution of 50 voxels in every direction, the scale factor is 40 m. Every voxel thus accounts for $40\,\text{m} \times 40\,\text{m} \times 40\,\text{m} = 64\,000\,\text{m}^3$ in volume. It has to be noted that this direct approach to rescaling and calculating the volume requires the model to be computed in cubic voxels.

## Appendix B: Results data

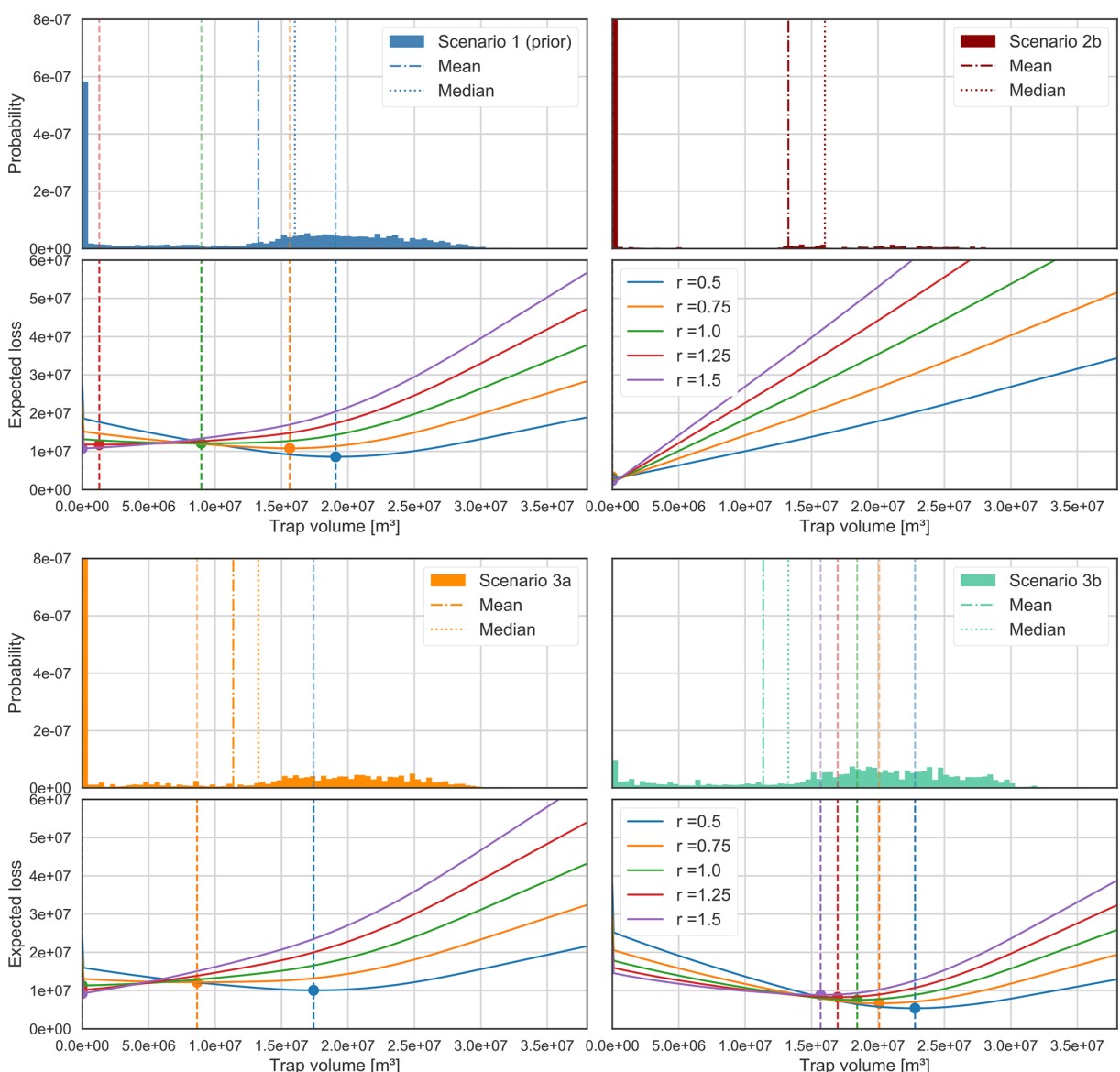

**Figure B1.** Posterior trap volume distributions and respective loss function realization plots for Scenarios 1 (prior), 2b, 3a, and 3b.

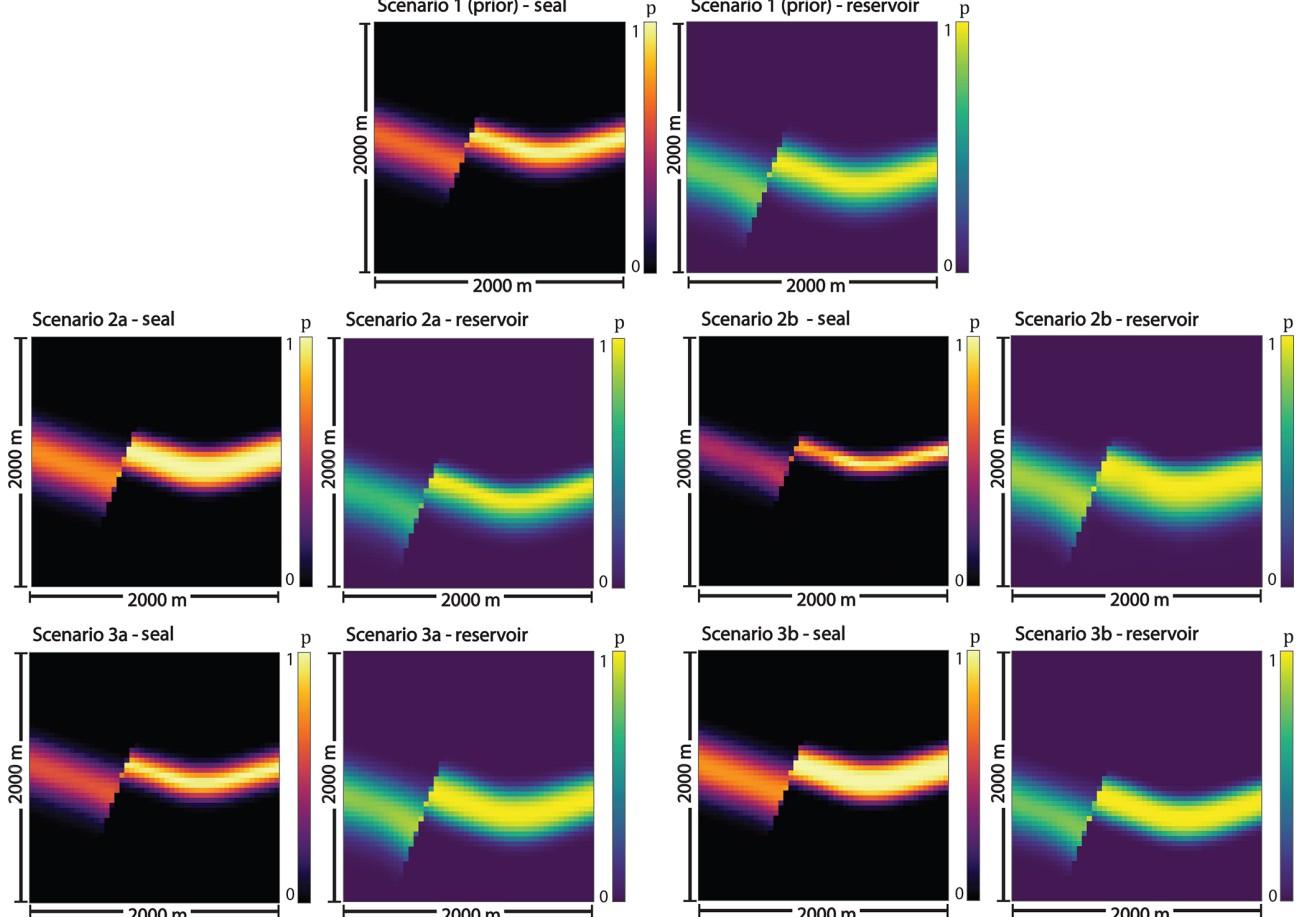

**Figure B2.** Probability field visualizations for Scenarios 1 to 3b.

**Table B1.** Decision results for all considered scenarios and each actor. Respective optimal estimates (decisions) are represented by $\hat{\theta}$, while $\Delta\hat{\theta}$ indicates posterior changes relative to the prior (Scenario 1) result. Expected losses are given by $l$, and changes relative to the prior by $\Delta l$.

| | | Decision makers | | | | |
| | | Risk friendly | | Risk neutral | Risk averse | |
| | | $r = 0.5$ | $r = 0.75$ | $r = 1.0$ | $r = 1.25$ | $r = 1.5$ |
| --- | --- | --- | --- | --- | --- | --- |
| Scenario 1 | $\hat{\theta}$ | 19 072 000.00 | 15 616 000.00 | 8 960 000.00 | 1 280 000.00 | 0.00 |
| Prior | $l$ | 8 582 112.55 | 10 785 632.54 | 12 100 484.80 | 11 759 772.46 | 10 763 671.94 |
| Scenario 2a | $\hat{\theta}$ | 22 528 000.00 | 19 712 000.00 | 17 920 000.00 | 16 448 000.00 | 15 232 000.00 |
| Thick seal | $\Delta\hat{\theta}$ | 3 456 000.00 | 4 096 000.00 | 8 960 000.00 | 15 168 000.00 | 15 232 000.00 |
| | $l$ | 5 387 582.96 | 6 654 239.73 | 7 544 384.00 | 8 220 155.30 | 8 776 678.80 |
| | $\Delta l$ | −3 194 529.59 | −4 131 392.81 | −4 556 100.80 | −3 539 617.16 | −1 986 993.14 |
| Scenario 2b | $\hat{\theta}$ | 0.00 | 0.00 | 0.00 | 0.00 | 0.00 |
| Thin seal | $\Delta\hat{\theta}$ | −19 072 000.00 | −15 616 000.00 | −8 960 000.00 | −1 280 000.00 | 0.00 |
| | $l$ | 2 743 719.13 | 2 240 237.29 | 1 940 102.40 | 1 735 280.34 | 1 584 086.98 |
| | $\Delta l$ | −5 838 393.42 | −8 545 395.25 | −10 160 382.40 | −10 024 492.12 | −9 179 584.96 |
| Scenario 3a | $\hat{\theta}$ | 17 408 000 | 8 640 000 | 0 | 0 | 0 |
| Thick reservoir | $\Delta\hat{\theta}$ | −1 664 000.00 | −6 976 000.00 | −8 960 000.00 | −1 280 000.00 | 0.00 |
| | $l$ | 10 073 515.53 | 12 159 993.48 | 11 319 609.6 | 10 124 566.62 | 9 242 422.54 |
| | $\Delta l$ | 1 491 402.98 | 1 374 360.94 | −780 875.20 | −1 635 205.84 | −1 521 249.40 |
| Scenario 3c | $\hat{\theta}$ | 22 784 000.00 | 20 096 000.00 | 18 432 000.00 | 16 960 000.00 | 15 680 000.00 |
| Thick reservoir and seal | $\Delta\hat{\theta}$ | 3 712 000.00 | 4 480 000.00 | 9 472 000.00 | 15 680 000.00 | 15 680 000.00 |
| | $l$ | 5 380 782.45 | 6 658 861.07 | 7 551 644.80 | 8 278 631.71 | 8 857 405.68 |
| | $\Delta l$ | −3 201 330.10 | −4 126 771.47 | −4 548 840.00 | −3 481 140.75 | −1 906 266.26 |
| Scenario 4a | $\hat{\theta}$ | 19 264 000.00 | 15 744 000.00 | 0.00 | 0.00 | 0.00 |
| Near-critical SSF | $\Delta\hat{\theta}$ | 192 000.00 | 128 000.00 | −8 960 000.00 | −1 280 000.00 | 0.00 |
| | $l$ | 8 959 284.13 | 11 533 073.67 | 13 250 828.80 | 11 851 901.58 | 10 819 256.41 |
| | $\Delta l$ | 377 171.58 | 747 441.13 | 1 150 344.00 | 92 129.12 | 55 584.47 |
| Scenario 4b | $\hat{\theta}$ | 23 040 000.00 | 20 992 000.00 | 19 584 000.00 | 18 496 000.00 | 17 664 000.00 |
| Reliable SSF and thick seal | $\Delta\hat{\theta}$ | 3 968 000.00 | 5 376 000.00 | 10 624 000.00 | 17 216 000.00 | 17 664 000.00 |
| | $l$ | 4 112 858.01 | 4 964 529.37 | 5 513 651.20 | 5 929 335.97 | 6 245 426.13 |
| | $\Delta l$ | −4 469 254.54 | −5 821 103.17 | −6 586 833.60 | −5 830 436.49 | −4 518 245.81 |

**Table B2.** Occurrence rate of trap control mechanisms in percent for each information scenario.

| | 1 – Anticlinal spill | 2 – Leak to reservoir | 3 – Leak to overlying | 4 – Stratigraphic breach | 5 – Unclear |
| --- | --- | --- | --- | --- | --- |
| Scenario 1 | 51.47 | 25.11 | 12.36 | 10.56 | 0.5 |
| Scenario 2a | 63.1 | 35.8 | 0.41 | 0.49 | 0.2 |
| Scenario 2b | 10.04 | 1.53 | 20.82 | 65.51 | 2.1 |
| Scenario 3a | 41.99 | 23.21 | 23.06 | 11.38 | 0.36 |
| Scenario 3b | 61.86 | 36.59 | 0.53 | 1.02 | 0 |
| Scenario 4a | 66.4 | 0.01 | 33.59 | 0 | 0 |
| Scenario 4b | 79.45 | 20.55 | 0 | 0 | 0 |

**Appendix C: MCMC convergence**

# Scenario 2a: Geweke plots

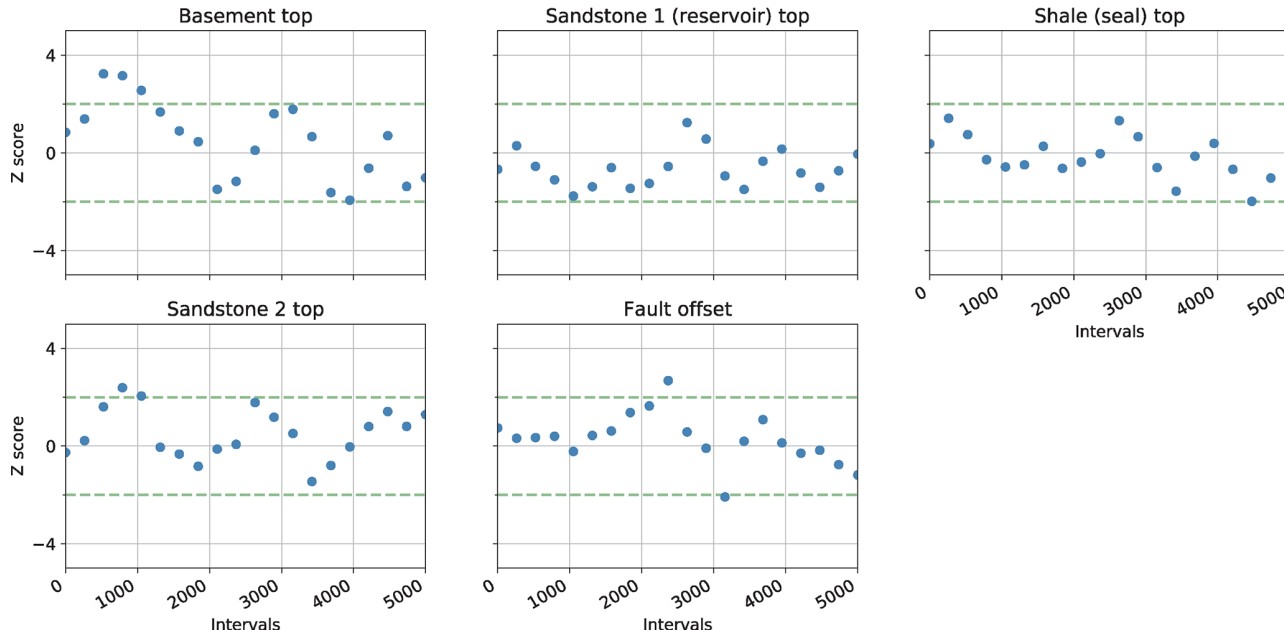

# Scenario 2b: Geweke plots

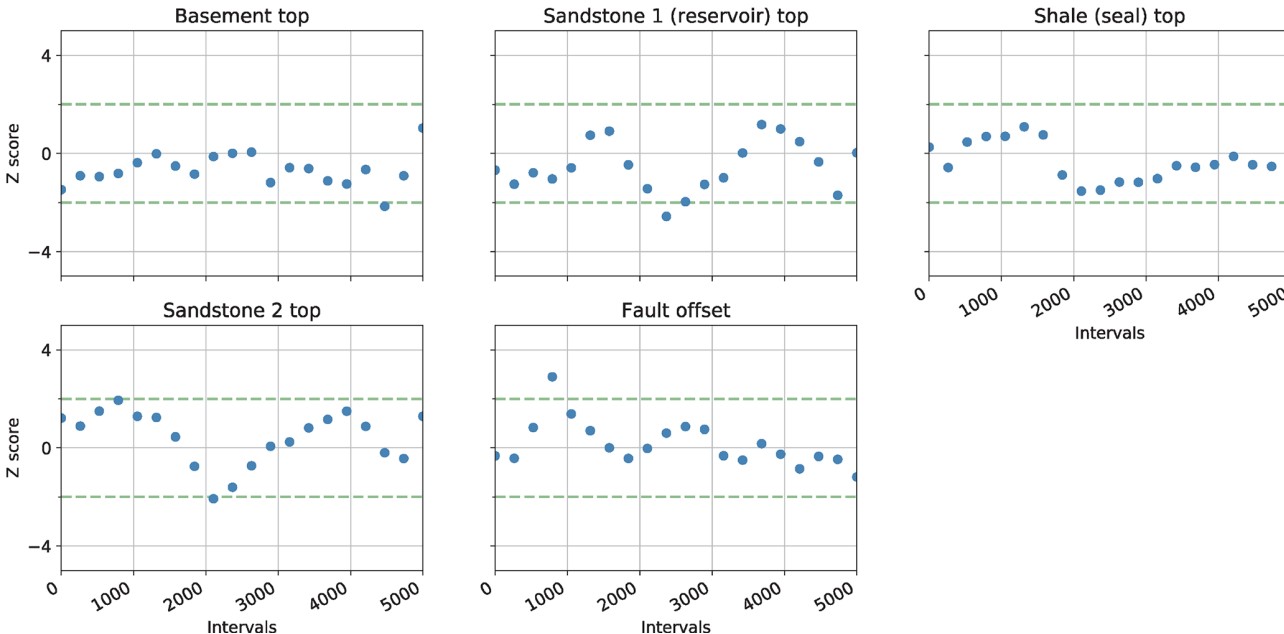

**Figure C1.**

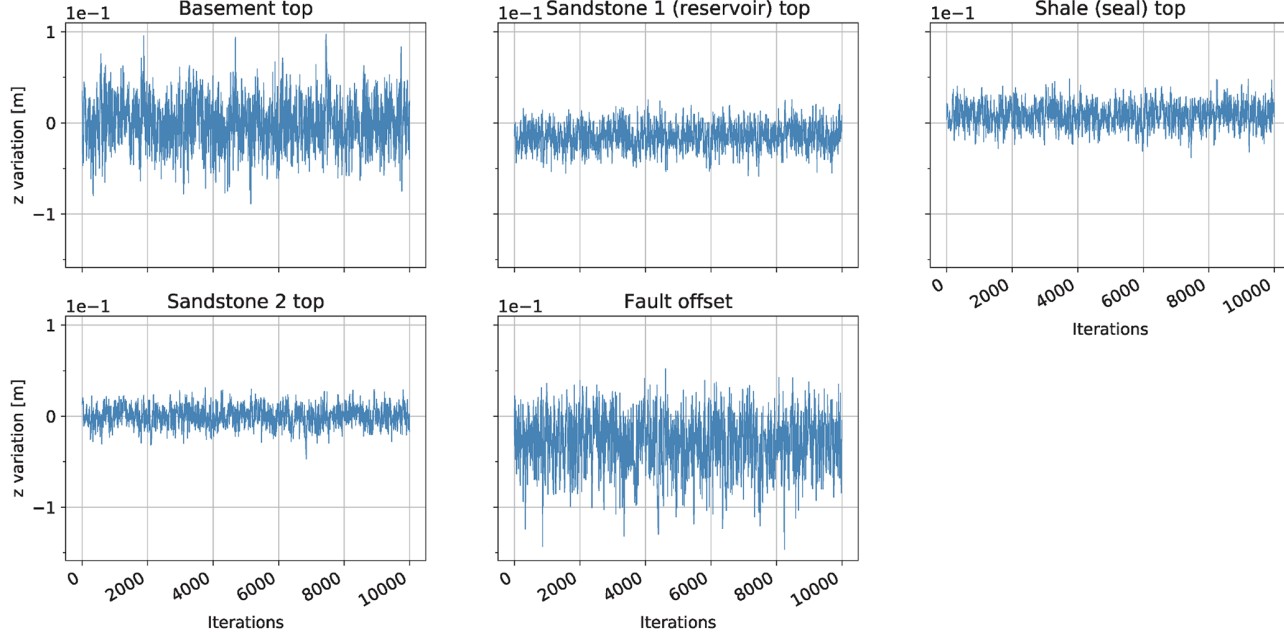

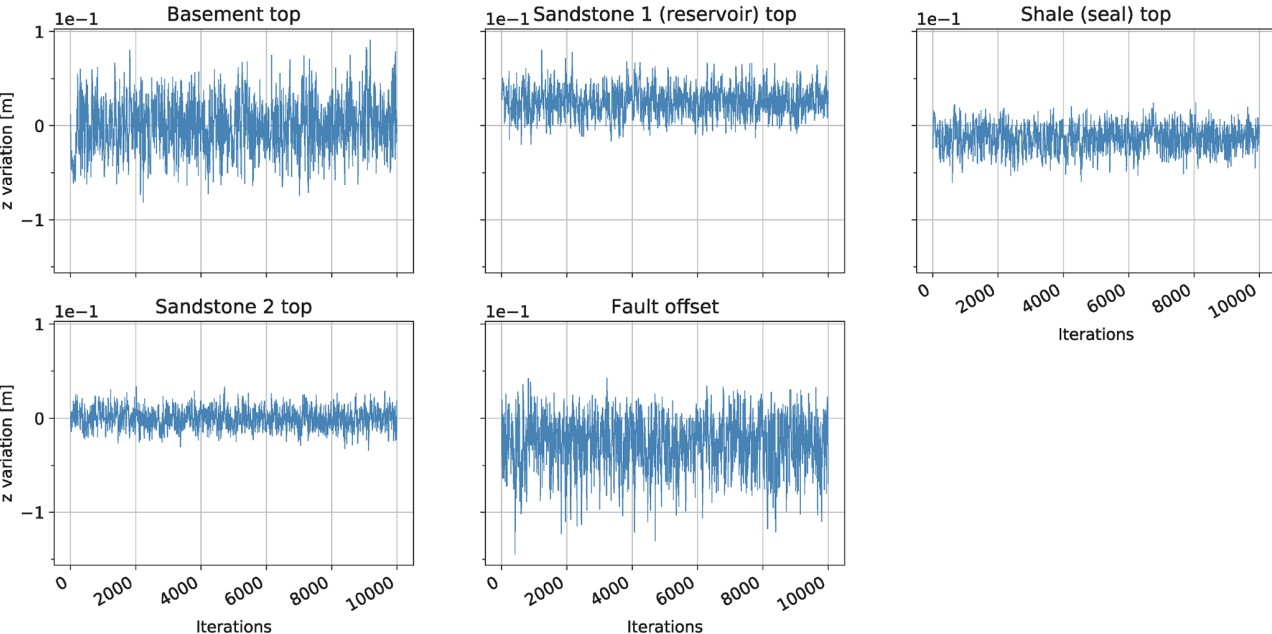

**Figure C1.**

# Scenario 3a: Geweke plots

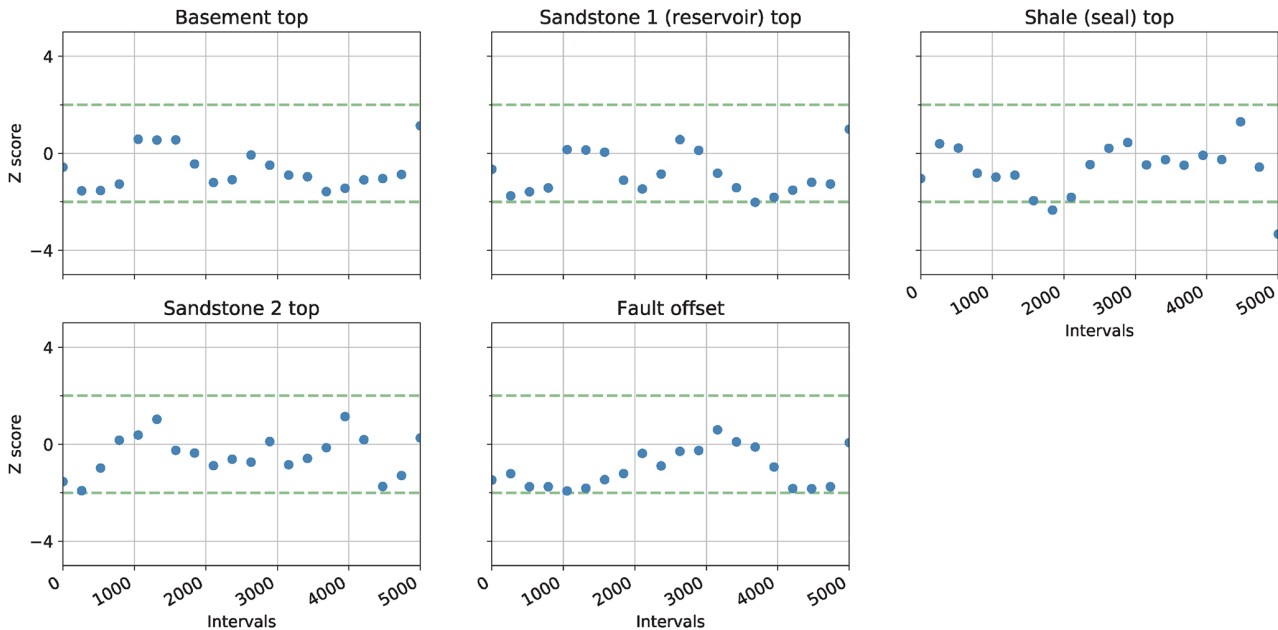

# Scenario 3b: Geweke plots

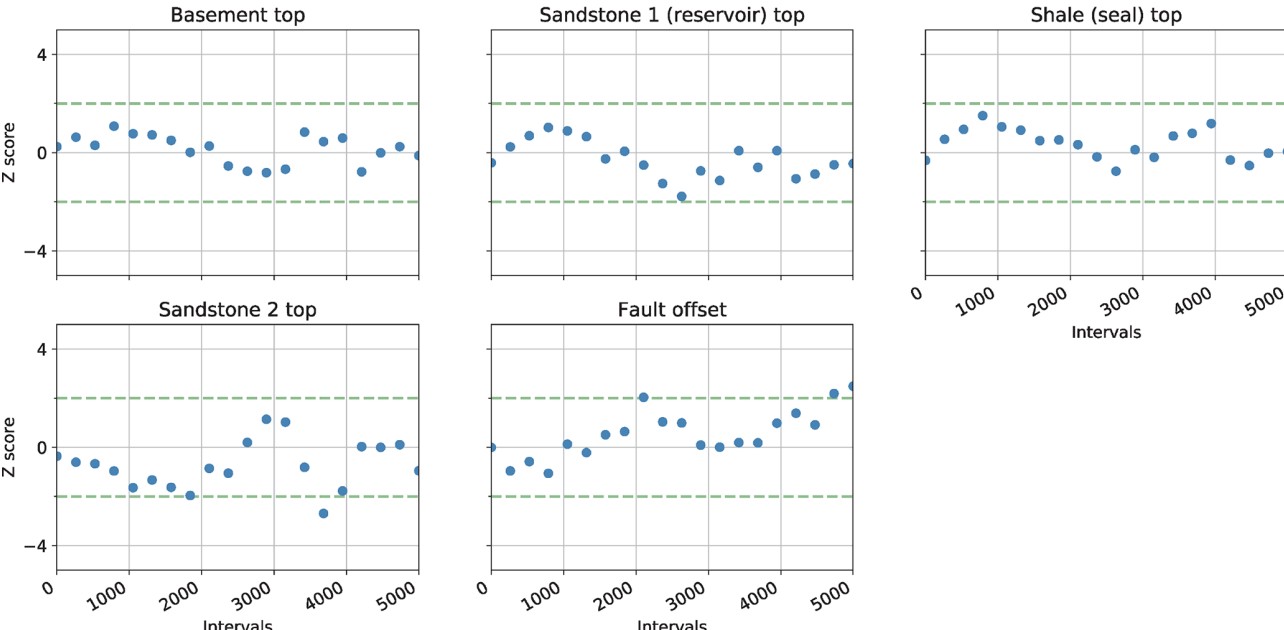

**Figure C1.**

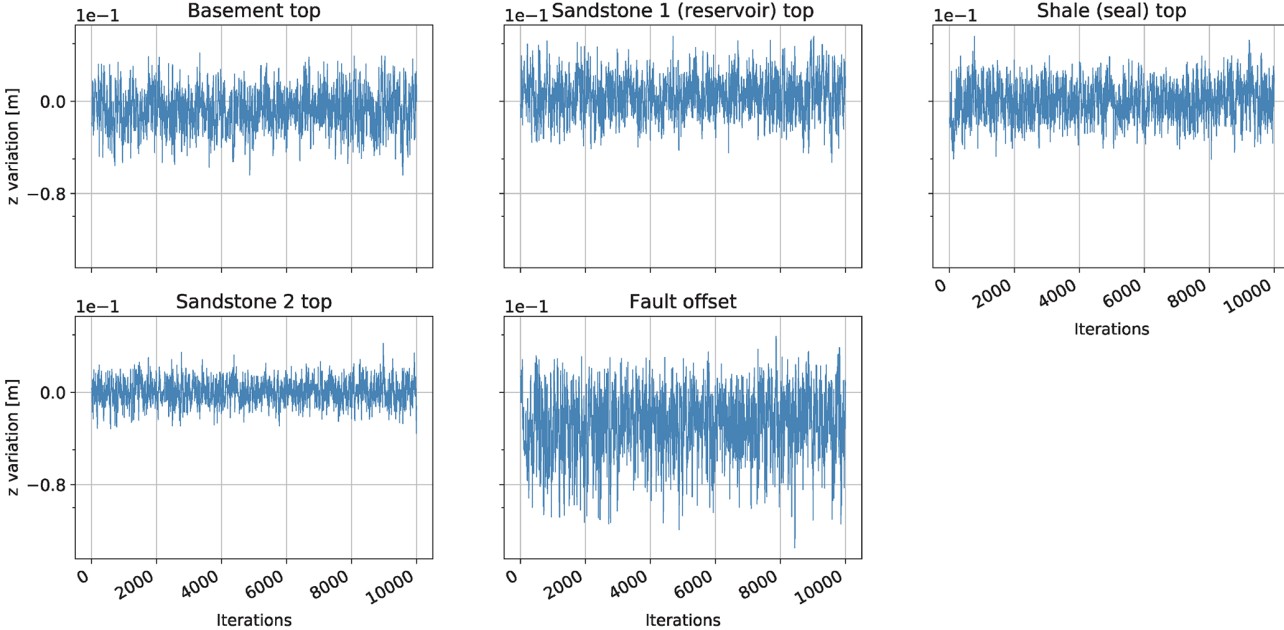

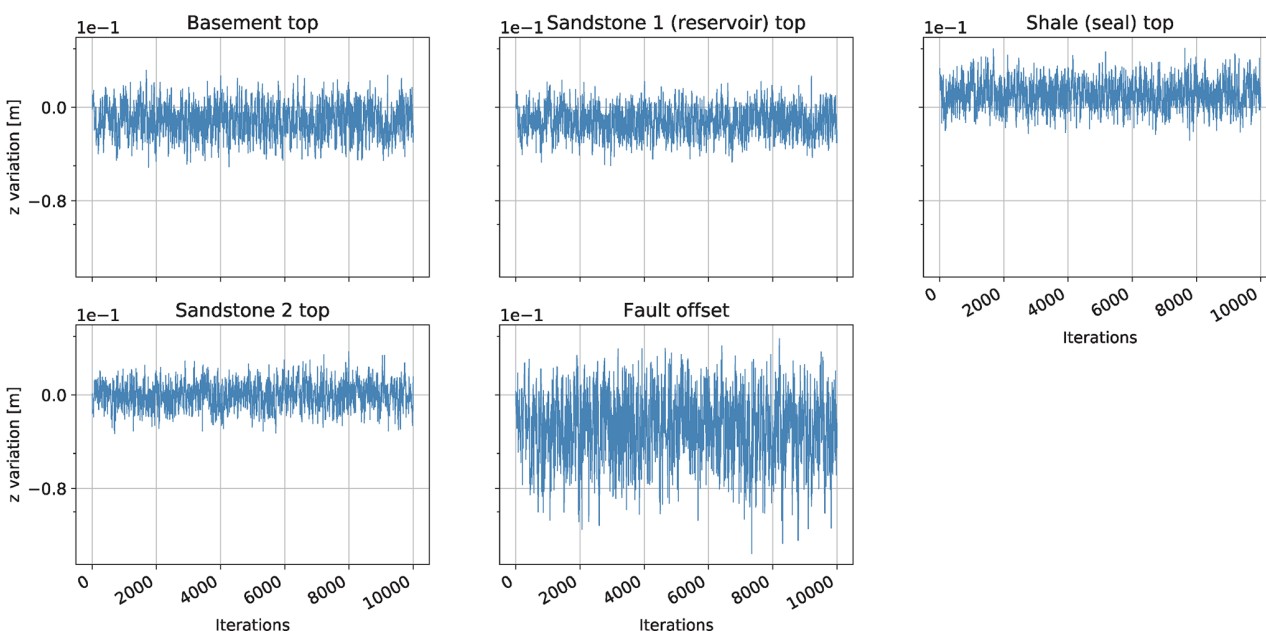

**Figure C1.** Geweke plots and traces for Scenarios 2a to 3b.

# Scenario 4a: Geweke plots

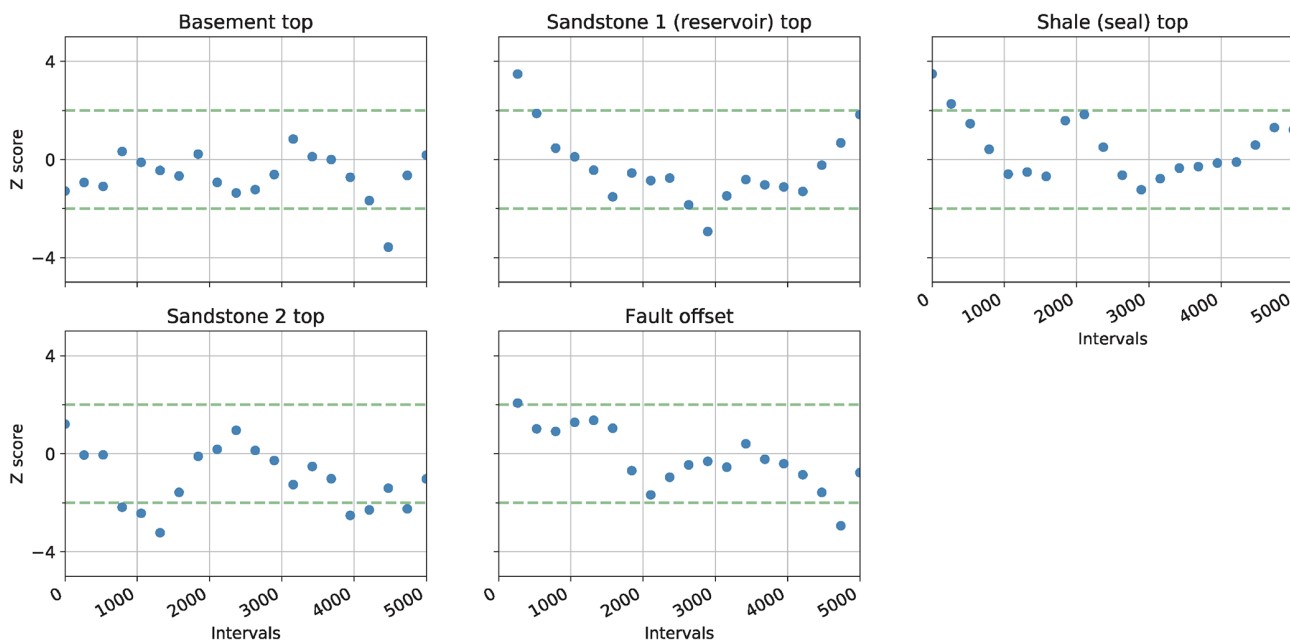

# Scenario 4b: Geweke plots

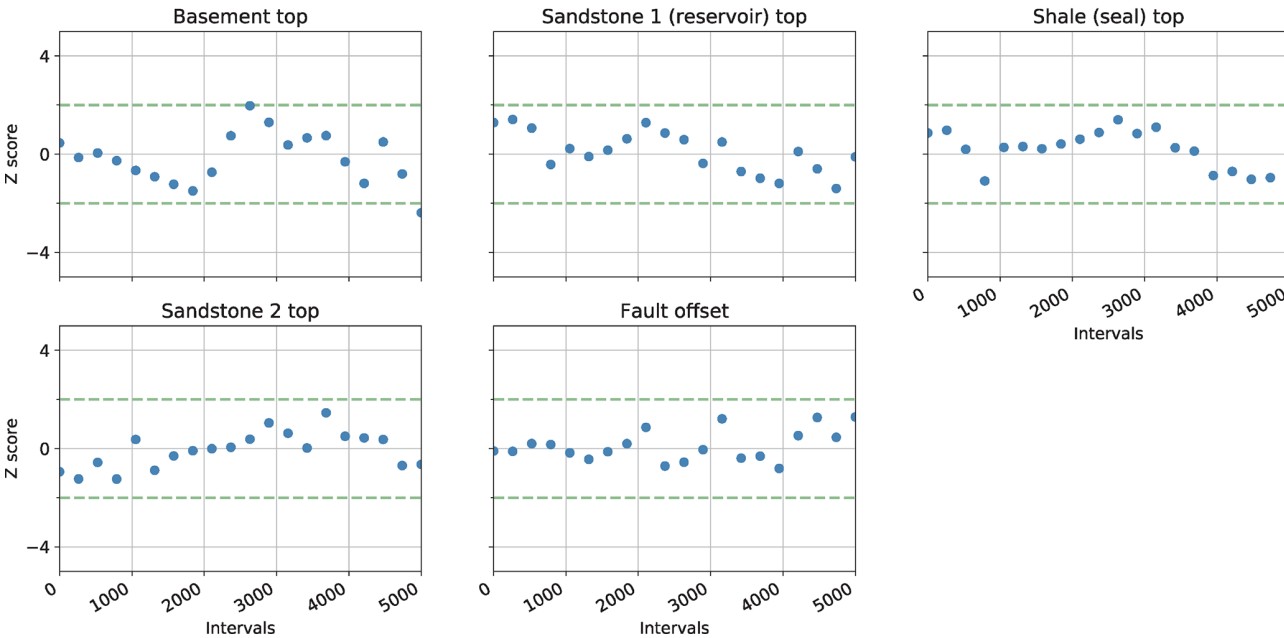

**Figure C2.**

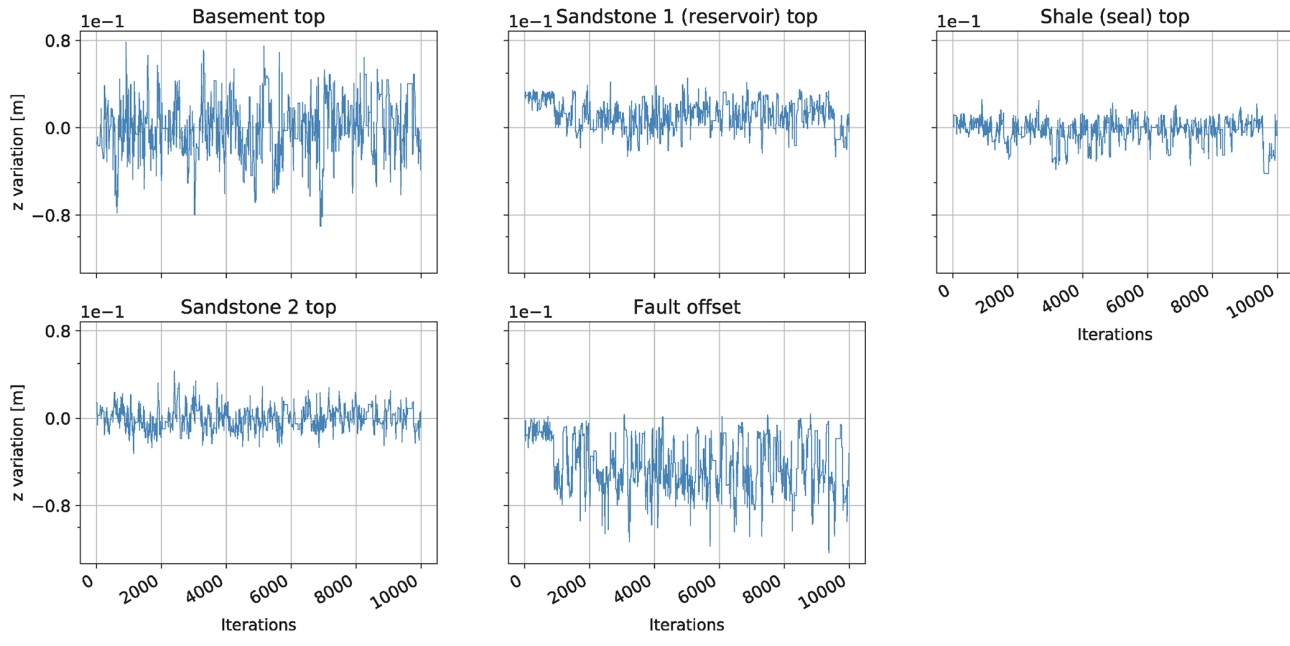

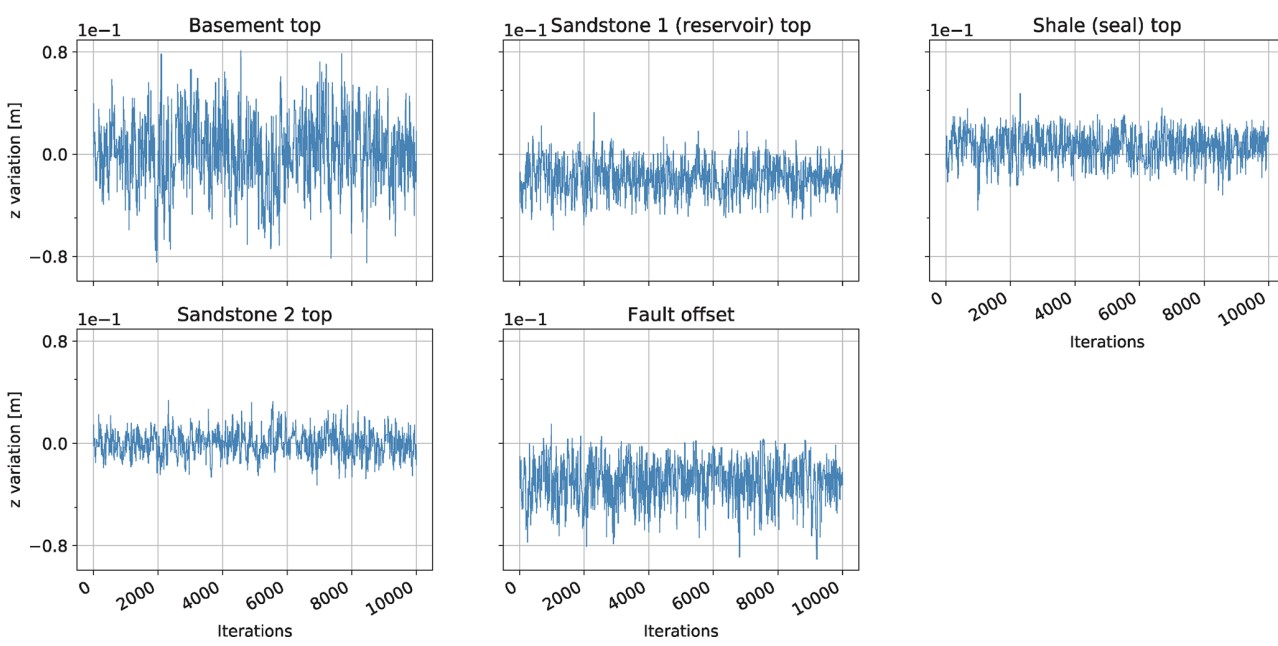

**Figure C2.** Geweke plots and traces for Scenarios 4a and 4b.

*Author contributions.* .TS4

*Competing interests.* The authors declare that they have no conflict of interest.

*Special issue statement.* This article is part of the special issue "Understanding the unknowns: the impact of uncertainty in the geosciences". It is a result of the EGU General Assembly 2018, Vienna, Austria, 8–13 April 2018.

OR

This article is part of the special issue "Understanding the unknowns: the impact of uncertainty in the geosciences". It is not associated with a conference.TS5

*Acknowledgements.* We would like to thank Cameron Davidson-Pilon for his comprehensive, free introduction into Bayesian methods, which inspired parts of this research. Special thanks to Alexander Schaaf for helping with 3-D visualizations.

*Review statement.* This paper was edited by Lucia Perez-Diaz and reviewed by two anonymous referees.

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

## Remarks from the language copy-editor

CE1    Please note that Figs. C1 and C2 were edited during copy-editing (adjusted for hyphenation only in accordance with our house standards). Please review the figure content carefully.

CE2    Please check this change. "Affine" seems to be used strictly in a mathematical sense: https://www.lexico.com/en/definition/affine.

## Remarks from the typesetter

TS1    The composition of Figs. 1, 3–8, and 10–12 has been adjusted to our standards.

TS2    Please confirm the change of * to · as a multiplication sign.

TS3    Please confirm.

TS4    Please note that the section "Author contributions" is mandatory.

TS5    Please decide which statement fits your context.

TS6    Please provide place of publication.

TS7    Please provide place of publication.

TS8    Please provide place of publication.

TS9    Please provide publisher and place of publication.

TS10    Please provide the volume.

TS11    Please provide place of publication.

TS12    Please list the names of all authors.

TS13    Please provide place of publication.

TS14    Please provide place of publication.

TS15    Please provide place of publication.

TS16    Please provide publisher and place of publication.

TS17    Please list the names of all authors.

TS18    Please list the names of all authors.

TS19    Please provide place of publication.

TS20    Please list the names of all authors.

TS21    Please provide place of publication.

TS22    Please provide place of publication.

TS23    Please list the names of all authors.

TS24    Please provide place of publication.

TS25    Please list the names of all authors.

TS26    Please provide the page range.

TS27    Please confirm.

TS28    Please list the names of all authors.

TS29    Please provide the date and location of the conference.

TS30    Please provide publisher and place of publication.

TS31    Please provide publisher and place of publication.

TS32    Please provide the page range or article number.

TS33    Please list the names of all authors.

TS34    Please provide place of publication.