# Peer review of "Actors, actions and uncertainties: Optimizing decision making based on 3-D structural geological models"

_Solid Earth, 2019_

## Short Comment (SC1) · 4 Apr 2019

This is a good paper with a focus on practical use of uncertainty propagation in implicit 3D geological modelling. The community indeed deserves more than mere uncertainty estimates and expects tools to extract useful knowledge from said estimates. Following risk assessment is risk mitigation.

The paper puts emphasis on the topological aspect of Monte Carlo Uncertainty Estimation (MCUE). That is, in MCUE the hypothetical opening and closing of traps becomes a discrete problem affected by piece-wise non-linearity.

[Figure]

This problem is difficult, although by no means it constitutes a limitation of the method and actually underlines one of its greatest strengths: contrary to analytical, simulated or inverse uncertainty propagation methods, the plausible models used in MCUE as intermediary steps are actually built and available for pre and post-analysis.

In this case, simple first order topological analysis methods applied to plausible models are demonstrated to be of potential use in oil & gas exploration and reserve estimation. One could regret the absence of a real case study. As the sampling is limited to formation thicknesses and fault offset, the MCUE process is also very unlikely to satisfactorily express the geometrical variability upon which the proposed method relies.

It is quite unsettling that the paper appears to inaccurately present these methods as new and unheard of (Winter 1994, Clementini 1997). Moreover, topological analysis applied to 3D geological modelling is fairly common (Deutsch 1998, Pellerin 2015). Topological analysis applied to MCUE methods is known (Thiele 2016, Pakyuz-Charrier 2018). The topic of topological analysis combined with risk estimation for oil & gas reservoir estimation has been covered too (Li 2012, Bazaikin 2013).

The work done is commendable and, in my opinion, is worth publishing. On a more personal note, I am pleased to witness other groups take on the topological aspect of MCUE.

Bazaikin, Y. V., Baikov, V. A., Taimanov, I. A., & Yakovlev, A. A. (2013). Numerical analysis of topological characteristics of three-dimensional geological models of oil and gas fields. arXiv preprint arXiv:1302.6885. Clementini, E., & Di Felice, P. (1997). Approximate topological relations. International journal of approximate reasoning, 16(2), 173-204. Deutsch, C. V. (1998). Fortran programs for calculating connectivity of three-dimensional numerical models and for ranking multiple realizations. Computers & Geosciences, 24(1), 69-76. Li, S., Deutsch, C. V., & Si, J. (2012, June). Ranking geostatistical reservoir models with modified connected hydrocarbon volume. In Ninth International Geostatistics Congress (pp. 11-15). Pakyuz-Charrier, E. J. (2018). Uncertainty

is an Asset: Monte Carlo simulation for uncertainty estimation in implicit 3D geological modelling. https://doi.org/10.26182/5c5b9c2bcf738 Pellerin, J., Caumon, G., Julio, C., Mejia-Herrera, P., & Botella, A. (2015). Elements for measuring the complexity of 3D structural models: Connectivity and geometry. Computers & Geosciences, 76, 130-140. Thiele, S. T., Jessell, M. W., Lindsay, M., Ogarko, V., Wellmann, J. F., & Pakyuz-Charrier, E. (2016). The topology of geology 1: Topological analysis. Journal of Structural Geology, 91, 27-38. Thiele, S. T., Jessell, M. W., Lindsay, M., Wellmann, J. F., & Pakyuz-Charrier, E. (2016). The topology of geology 2: Topological uncertainty. Journal of Structural Geology, 91, 74-87. Winter, S. (1994, August). Uncertainty of topological relations in GIS. In ISPRS Commission III Symposium: Spatial Information from Digital Photogrammetry and Computer Vision (Vol. 2357, pp. 924-931). International Society for Optics and Photonics.

---

## Short Comment (SC2) · 11 Apr 2019

Dear Stamm et al.,

I think that this submission tackles a challenge that has eluded many practitioners so far by bridging geological uncertainty and decision-making in a quantitative way. I have read the manuscript with interest, and there are a few comments that I would like to make where it relates to my expertise or experience.

I feel that the introduction might not review existing work in sufficient depth. Previous studies use Monte-Carlo simulations using geological measurement uncertainty

to produce series of models that respect geological plausibility filters and topological laws while honouring said geological measurements, subject to their uncertainty levels. Some of the previous works I refer to are cited in a reference given in the text (Wellmann and Caumon (2018)), but I think that they deserve to be mentioned directly in your manuscript. The review of geological uncertainty highlighting how it is addressed (or, actually, not always really addressed) at the different stages of the workflow by Jessell et al. (2018) also has its place in the introduction. I also think that works by Schneeberger et al. (2017) and Schweizer et al. (2017) may be contributions that relate to the topics covered here.

My argument is that since uncertainty estimation in geological modelling is one of the essential building blocks of this manuscript, the review of the literature should not be as restrictive. It covers a field of active research and should incorporate other works following ideas from Wellmann et al. (2010) such as Pakyuz-Charrier et al. (2018a), who perform Monte-Carlo simulations following the general idea developed in the previous paragraph using a 'Monte Carlo Uncertainty Estimator'. Besides, their work also utilise the implicit modelling framework of Calcagno et al. (2008) and considers the lithologies' apparition frequency. On another note, since the drilling of wells is the one key to successful prospect development in both conventional and shale hydrocarbon contexts (maybe to the exception of bitumen sands), it might worth mentioning that uncertainty in well trajectory and measurements has been recognised as an issue, and addressed using the same framework relying on Monte-Carlo simulations (Pakyuz-Charrier et al. (2018b)). One last thing about probabilistic geological modelling. I think that this area of the geosciences has seen more developments in hard rock scenarios than in basin studies, which should be accounted for when targeting oil and gas studies.

Another aspect of oil and gas exploration which is essential to exploration success is seismic modelling. P10 and P90, which you rightly mention as being commonly used for uncertainty evaluation and decision making, are, more often than not, derived at least in part from such modelling. The field of stochastic/probabilistic seismic inversion has been gaining traction in oil and gas E&P industry for some time now (many examples are available in the literature), while uncertainty in seismic interpretation is gaining interest (see for instance Bond (2015), Alcalde et al. (2017), and Solid Earth Discussion manuscripts by Schaaf and Bond (2019) and Bárbara et al. (2019), both of which postdate your submission). The same applies, to a certain extent, to the study of uncertainty and geological risk management using electromagnetic methods in hydrocarbon frontier exploration (see for instance Meju (2019) and Baltar and Roth (2013)).

Last, practical applications and underlying hypotheses. In deriving 'geological' models from the interpretation of seismic data, which is often constrained by well information, geoscientists need to consider the lateral and vertical resolution of their imaging. In such cases, uncertainty is heteroscedastic and is a function of a number of parameters such as data fold, frequencies, S/N ratio, etc. It is not trivial to estimate, in particular knowing that dipping structures are much more uncertain than horizontal ones. In short, I think that the role that geophysics plays in imaging, uncertainty and prospect evaluation in the oil and gas E&P should at least be stated briefly, and not completely absent from the text as it is currently the case, which I find a little bit surprising. Someone might wonder what data are used to derive the geometry and location of structural traps in this kind of conventional play, and even the predefined uncertainty levels in positioning, if not mostly seismic and borehole data.

Notwithstanding the comments above, I find this submission interesting, and I think that it is worth a journal publication. I will be looking forward to seeing the revised version of the manuscript.

I hope that my criticism will be helpful and constructive,

Best regards,

Jeremie Giraud Centre for Exploration Targeting, School of Earth Sciences, University of Western Australia.

References:

Alcalde, J., C. E. Bond, G. Johnson, J. F. Ellis, and R. W. H. Butler, 2017, Impact of seismic image quality on fault interpretation uncertainty: GSA Today.

Baltar, D., and F. Roth, 2013, Reserves estimation methods for prospect evaluation with 3D CSEM data: First Break.

Bárbara, C. P., P. Cabello, A. Bouche, I. Aarnes, C. Gordillo, O. Ferrer, M. Roma, and P. Arbués, 2019, Quantifying the impact of the structural uncertainty on the gross rock volume in the Lubina and

Montanazo oil fields (Western Mediterranean): Solid Earth Discussions, 1–36.

Bond, C. E., 2015, Uncertainty in structural interpretation: Lessons to be learnt: Journal of Structural Geology, 74, 185–200.

Calcagno, P., J. P. Chilès, G. Courrioux, and A. Guillen, 2008, Geological modelling from field data and geological knowledge. Part I. Modelling method coupling 3D potential-field interpolation and geological rules: Physics of the Earth and Planetary Interiors, 171, 147–157.

Jessell, M., E. Pakyuz-charrier, M. Lindsay, J. Giraud, and E. de Kemp, 2018, Assessing and Mitigating Uncertainty in Three-Dimensional Geologic Models in Contrasting Geologic Scenarios: , 63–74.

Meju, M. A., 2019, A simple geologic risk-tailored 3D controlled-source electromagnetic multiattribute analysis and quantitative interpretation approach: GEOPHYSICS, 84, E155–E171.

Pakyuz-Charrier, E., M. Lindsay, V. Ogarko, J. Giraud, and M. Jessell, 2018a, Monte Carlo simulation for uncertainty estimation on structural data in implicit 3-D geological modeling, a guide for disturbance distribution selection and parameterization: Solid Earth, 9, 385–402.

Pakyuz-Charrier, E., J. Giraud, V. Ogarko, M. Lindsay, and M. Jessell, 2018b, Drillhole uncertainty propagation for three-dimensional geological modeling using Monte Carlo: Tectonophysics.

Schaaf, A., and C. E. Bond, 2019, Quantification of uncertainty in 3-D seismic interpretation: implications for deterministic and stochastic geomodelling and machine learning: Solid Earth Discussions, 1–18.

Schneeberger, R., M. D. La Varga, D. Egli, A. Berger, F. Kober, F. Wellmann, and M. Herwegh, 2017, Methods and uncertainty estimations of 3-D structural modelling in crystalline rocks : a case study: , 987–1002.

Schweizer, D., P. Blum, and C. Butscher, 2017, Uncertainty assessment in 3-D geological models of increasing complexity: Solid Earth, 8, 515–530.

Wellmann, F., and G. Caumon, 2018, 3-D Structural geological models: Concepts, methods, and uncertainties, in , 1–121.

Wellmann, J. F., F. G. Horowitz, E. Schill, and K. Regenauer-Lieb, 2010, Towards incorporating uncertainty of structural data in 3D geological inversion: Tectonophysics, 490, 141–151.
* * *

---

## Author Comment (AC1) · 17 Apr 2019

Dear Evren Pakyuz-Charrier,

Thank you very much for reading our manuscript and commenting on it. Considering some of your remarks, it seems to be the case that aspects of our manuscript can be misunderstood, if read in a certain way - especially with regards to the actual focus of this paper. Possibly, we did not make our intent sufficiently clear from the beginning, and we wish to correct this not only here, but also in a revised version of the manuscript.

As stated several times throughout our original text, we propose the use of custom

loss functions as a Bayesian decision-making tool that could be used in the context of geological modeling. This is what we regard as a novel approach, in this field at least. In no way did we mean to convey that we present Monte Carlo simulation, topological analysis, uncertainty quantification or any combination of these methods as something new. We have worked and published in this field for almost a decade (Wellmann et al., 2010) and are well aware of the existing literature. In fact, topological analysis in particular is absolutely not the focus of our work (the term is not even mentioned anywhere in the manuscript). To be sure, the algorithms for trap volume calculation we use are topology-related, but we merely use this as a basis to attain the maximum trap volume as an intermediate quality of interest. Our focus lies on the subsequent aspects of Bayesian decision theory, which could be applied to very different parameters and settings, disregarding topology. Of course, topological analyses would add interesting aspects to the sampling itself, and we are currently preparing a manuscript with more details - but this is not the context of the work presented here.

It is true, that a real case study would have been nice to examine, and this was acknowledged in our discussion section, so were the limitations regarding geometrical variability. While this is a good point to make, it does not affect the main aspect of this work, which is the evaluation of the method we employ for decision making. As we combined our method directly with probabilistic geological modeling approaches implemented in the software GemPy (de la Varga et al., 2019), one could readily apply the Bayesian decision making approach to more complex scenarios. We limited the study to a conceptual case of a typical trap structure, in order to provide an intuitive example that many researchers can easily relate to.

In addition, we would like to express our concern regarding the use of "Monte Carlo Uncertainty Estimation" (MCUE) as an actual term. We share the view of Prof. Caumon who, in the review of Pakyuz-Charrier et al., 2018 (available here: https://www.solid-earth-discuss.net/se-2017-115/se-2017-115-RC3.pdf), stated: "Monte Carlo simulation for uncertainty estimation" (as in the title) seems clearer to me than "Monte Carlo

simulation uncertainty estimation" (...). Therefore, I would recommend to replace MCUE by a more specific term (including in the paper's title)". We also consider that coining a widely used concept with a new term is potentially misleading. Furthermore, the few references that were primarily used to support its usage (Camacho et al 2015, Beven & Binley, 1992), do not mention the term MCUE.

Nevertheless, we acknowledge that you have highlighted some shortcomings on our side. We apparently did not communicate well enough the difference between the methods which we choose as an exemplary basis, and the ones we introduce as a new and possibly useful tool in this context. We will clarify the above points in the revised version of the manuscript. You also named some references that are certainly worth considering and that we will include in the revised version of our paper.

Beven, K., & Binley, A. (1992). The future of distributed models: model calibration and uncertainty prediction. Hydrological processes, 6(3), 279-298.

Camacho, R. A., Martin, J. L., McAnally, W., Díaz‐Ramirez, J., Rodriguez, H., Sucsy, P., & Zhang, S. (2015). A comparison of Bayesian methods for uncertainty analysis in hydraulic and hydrodynamic modeling. JAWRA Journal of the American Water Resources Association, 51(5), 1372-1393.

de la Varga, M., Schaaf, A., and Wellmann, F. (2019). GemPy 1.0: open-source stochastic geological modeling and inversion. Geosci. Model Dev., 12, 1-32, https://doi.org/10.5194/gmd-12-1-2019.

Pakyuz-Charrier, Evren & Lindsay, Mark & Ogarko, Vitaliy & Giraud, Jeremie & Jessell, Mark. (2018). Monte Carlo simulation for uncertainty estimation on structural data in implicit 3-D geological modeling, a guide for disturbance distribution selection and parameterization. Solid Earth, 9, 385-402, 10.5194/se-9-385-2018.

Wellmann, J. F., Horowitz, F. G., Schill, E., & Regenauer-Lieb, K. (2010). Towards incorporating uncertainty of structural data in 3D geological inversion. Tectonophysics,

490(3-4), 141-151, https://doi.org/10.1016/j.tecto.2010.04.022.

---

## Referee Comment (RC1) · Anonymous Referee #1 · 13 May 2019

This is an interesting paper, I think that the use of loss functions in this setting is a useful development and one with considerable potential, and I think that the paper is to be welcomed because of this. That said, the account of the statistical formulation of the author's model is very unclear and needs considerable improvement.

In section 2.2 the authors present their synthetic setting. This can be modelled, by an implicit modelling algorithm, from some set of input values. I assume that these input values are notional values of the $z$ coordinate for a particular contact at each of a set of locations $\{x, y\}$ in addition to notional values of dip direction and angle. The authors state that they can sample 'deviation values' (I assume they mean errors)

[Figure]

for these observations which are then added to the 'known' values. With one major caveat (below), the application of the implicit modelling algorithm to each of a series of 'observations' with stochastic error would generate a series of models of the underlying state of affairs. The caveat is that the authors give no indication of how they specify the joint distribution of errors, i.e. the joint distribution for the errors in $z$ at locations $x_1, y_1$ and $x_2, y_2$. One can assume only that they treat these errors as independent, but that is most implausible in any geological setting, and could have dramatic consequences for the resulting model output. There are several studies, including some cited in this paper, which have considered the spatial dependence of model errors, and the authors should give this careful consideration in revising their work.

The authors then attempt to set their work in a Bayesian context. However, they do not make this at all explicit, and it is very difficult to understand or assess what they have done. A Bayesian analysis requires some model, for which some set of parameters are unknown and are treated as random variables to reflect this uncertainty. Prior probability distributions express the modeller's subjective view, prior to examining the data under consideration, about the values of these parameters. Likelihood values can be computed for proposed values of the parameters, and this, possibly in conjunction with some sampling method such as MCMC, is then used to characterize the posterior distribution of the parameters, given the data. Posterior distributions, or samples from them, could then be used, for example, to estimate the expected value of a loss function under a particular decision rule.

While the paper starts with some very generalized statements about Bayesian methods (Section 2.4), we are nowhere given a sufficiently clear account of their model system and how it is expressed in Bayesian form. The jump from 2.4 to 2.4.1 is much too large. It is not possible for the reader to extract from section 2.4.1 a sufficiently clear account of what the authors have done to be able to reproduce it in a comparable setting, or to be confident that it has been done correctly. For example, the authors refer to 'layer thickness likelihoods'. I assume that the thickness of a layer is a model

parameter (assumed to be the same everywhere in space?). It is not made clear how the likelihood is defined, given a set of notional observations. Are the observations treated as independent? Is this reasonable? I would certainly be concerned if the thicknesses of the separate layers are treated as independent random variables. This does not strike me as implausible and, as assumptions go, it is likely to be a sensitive one in this particular context.

In short, I think that the authors need to start again, aiming at the simplest and clearest account they can give of their modelling framework so as to exemplify the loss function concept to best advantage. At present this gets lost in vague and hand-waving description.

A couple of minor points. On page 6 line 10 you refer to a negatively skew normal distribution of mean zero used to generate errors of the throw for your fault. One would expect such a distribution to have a median value larger than the mean, and there statement that most of the values produced are negative is curious. Please clarify.

Second, the statement of the loss functions from page 11 onward should be clarified. As stated Equations (4)–(7) are identical upto a constant value. I assume that you actually introduce the constants in (5) onward to introduce different degrees of asymmetry. This should be expressed in a rigorous way, indicating the conditions over which the constant is introduced. With some care a parsimonious but rigorous notation could be developed.

---

## Referee Comment (RC2) · Anonymous Referee #2 · 17 Jun 2019

This is an interesting and original contribution on simulating uncertainties and their impact on decision making in a structural hydrocarbon trap. Monte Carlo error propagations and Markov Chain Monte Carlo sampling are used to consider the probability of different trap volume models based on stated uncertainties. Loss functions are used to explore potential decision making scenarios for high-risk and low-risk users. The technique has great potential to be used, with addition of more parameters, for hydrocarbon exploration and other geoscience applications.

The main strengths of the paper are that it is tightly focused on a specific and important problem, it builds up the methods and results in logical step-wise sections, and it

contains useful and clear figures to clarify key inputs/findings. The main weaknesses are that it is written in a fairly inaccessible way, with key elements and some assumptions undefined, and that it does not look beyond the broader applications beyond the specific one used.

There a few ways that it could be made more accessible to a non-specialist in decision analysis (discussed below) and some general editing would shorten the text and increase its clarity.

General: For reviewing, it would be much better to have continuous line numbers to refer to rather than numbering resetting back to 1 on each page, also every line rather than every 5th line should be numbered.

Lots of instances of unnecessary modifiers e.g. line 10 (P1), line 3-4 (P4), convoluted sentences (e.g. line 2-4, P9) and redundant words (e.g. line 20 P12, line 4, P19) make the text difficult to access. Many of these could be drastically simplified, making the text shorter and more accessible.

The word 'actor' is used throughout and in the title. It would be good to define the term in the context used early on. I see that it is commonly used in the decision sciences as a synonym for the more widely-understood term 'decision maker' or 'user', and seems to represent the human element of the process. However, in line 2 (P9) the phrase 'the case of an actor or decision maker' is used, suggesting that the two are different – perhaps actor has a very specific meaning here, which should be made clear. In line 11 (P9), for example, the 'actor' would like to know the trap volume. Use of actor in this context seems obscure, and would be better replaced with a common word representing the human element in decision analysis like worker/geologist/company/user. Line 15 (P17) reports observations of the behaviour of the modelled actors – reflecting that actors are a non-human element of the modelling (e.g. Fig. 3). Please clarify.

Several other key concepts, terms and abbreviations are used throughout, but not all are clearly defined. It would be beneficial to place the work in the framework of such

definitions.

The use of the hydrocarbon sector as an application for geological modelling is sensible. However, no other options are mentioned. What might other sectors be, and what modelling problems could be solved in the same way (perhaps things like nuclear waste disposal, landslide susceptibility)?

Throughout, volumetrics are discussed in the sense of 'trap volume' – which is appropriate and supported by the parameters used. However, there is one instance of the term 'reservoir volume', and there is consideration, in section 2.3, of OOIP. This indicates that actual hydrocarbon reserves are a key outcome of the analysis. But converting from a trap volume to a reserve volume of course involves extra parameters such as porosity, net:gross ratios, water saturation etc. These factors and their uncertainties may be as important or more important than the overall trap volume. While it is reasonable that other factors aren't included in the present modelling, it would be beneficial to mention them, justify their omission, and perhaps consider how they might be integrated in a future iteration of the model. I think they are alluded to in Fig 7 (reservoir and recovery parameters) but these should be made explicit, given the geologic focus of the topic.

Lines 6-8 (P6). Is there actually an independent uncertainty related to fault offset? Since that parameter can only be inferred indirectly via stratigraphic surface picks, I would argue that there is no additional uncertainty on either the hangingwall or footwall beyond what has already been accounted for by the surface uncertainty (which is the sole observational basis for fault offset). It would be useful to see a short descriptive justification for including this additional uncertainty. The significance of this parameter is clear in Fig. 5, where the smaller probabilities of hangingwall seal and reservoir result (I think) from the additional uncertainty applied. If this extra uncertainty is not justified (I think it is not), then it places the subsequent results in doubt.

Line 14 (P6) OOIP/OOIG – presumably this should be OOIP/OGIP instead?

Line 25 (P6). SSF is not defined. I suspect it may be 'shale smear factor' (used without the abbreviation in line 19 (P18)). Please define accordingly.

Line 18 (P13). 'Low but positive volumes' – is a negative volume possible/meaningful? If not, simply use 'low volumes'.

A few minor typos throughout (e.g. line 24 (P18) should read: an individual's..., line 29 (P18) should read: to what extent...). Please check and amend generally.

---

## Author Comment (AC2) · 13 Jul 2019

**RC1 (Anonymous Referee # 1)**

*This is an interesting paper, I think that the use of loss functions in this setting is a useful development and one with considerable potential, and I think that the paper is to be welcomed because of this.*

- We thank the reviewer for this motivating content on the topic itself.

[Figure]

*That said, the account of the statistical formulation of the author's model is very unclear and needs considerable improvement.*

*In section 2.2 the authors present their synthetic setting. This can be modelled, by an implicit modelling algorithm, from some set of input values. I assume that these input values are notional values of the z coordinate for a particular contact at each of a set of locations{ x, y} in addition to notional values of dip direction and angle.The authors state that they can sample 'deviation values' (I assume they mean errors) for these observations which are then added to the 'known' values. With one major caveat (below), the application of the implicit modelling algorithm to each of a series of 'observations' with stochastic error would generate a series of models of the underlying state of affairs. The caveat is that the authors give no indication of how they specify the joint distribution of errors, i.e. the joint distribution for the errors in z at locations x1, y1 and x2, y2. One can assume only that they treat these errors as independent, but that is most implausible in any geological setting, and could have dramatic consequences for the resulting model output. There are several studies, including some cited in this paper, which have considered the spatial dependence of model errors, and the authors should give this careful consideration in revising their work.*

- The reviewer is right in his criticism of the description of our probabilistic model. We did not focus our attention too much on this aspect, as the probabilistic model is not the central point of the paper (see below).

*The authors then attempt to set their work in a Bayesian context. However, they do not make this at all explicit, and it is very difficult to understand or assess what they have done. A Bayesian analysis requires some model, for which some set of parameters are unknown and are treated as random variables to reflect this uncertainty. Prior*

*probability distributions express the modeller's subjective view, prior to examining the data under consideration, about the values of these parameters. Likelihood values can be computed for proposed values of the parameters, and this, possibly in conjunction with some sampling method such as MCMC, is then used to characterize the posterior distribution of the parameters, given the data. Posterior distributions, or samples from them, could then be used, for example, to estimate the expected value of a loss function under a particular decision rule.*

- This is exactly what we attempt to show in this paper - and we realize from the comment that our description in this regard was not sufficiently clear (see below).

*While the paper starts with some very generalized statements about Bayesian methods (Section 2.4), we are nowhere given a sufficiently clear account of their model system and how it is expressed in Bayesian form. The jump from 2.4 to 2.4.1 is much too large. It is not possible for the reader to extract from section 2.4.1 a sufficiently clear account of what the authors have done to be able to reproduce it in a comparable setting, or to be confident that it has been done correctly. For example, the authors refer to 'layer thickness likelihoods'. I assume that the thickness of a layer is a model parameter (assumed to be the same everywhere in space?). It is not made clear how the likelihood is defined, given a set of notional observations. Are the observations treated as independent? Is this reasonable? I would certainly be concerned if the thicknesses of the separate layers are treated as independent random variables. This does not strike me as implausible and, as assumptions go, it is likely to be a sensitive one in this particular context.*

- We did not go into too much detail about Bayesian inference, as it is not of central importance in this work. The loss function approach should equally be applicable in a general probabilistic context, using results from common Monte Carlo simulation. Therefore, we are planning on omitting the application of Bayesian inference from the revised version of this paper, in order to avoid any confusion and focus on the loss function part. We will instead mention that suitable parameter correlations in an actual application could be obtained through Bayesian inference, and we will include appropriate references to existing work in this direction.

*In short, I think that the authors need to start again, aiming at the simplest and clearest account they can give of their modelling framework so as to exemplify the loss function concept to best advantage. At present this gets lost in vague and hand-waving description.*

- We acknowledge that the manuscript in its current form broaches too many subjects without explaining each of them in adequate detail. This makes it difficult to follow and understand the actual focus of this work, which is the conceptual application of the loss function approach for decision making in the context of geological exploration. As also suggested by you, in revision, we will streamline this work by limiting implementations to solely the most important methods and elaborations, and better defining the terms and approaches used. In fact, an earlier draft version of our manuscript included a simplified model with a more detailed description of the probabilistic decision workflow. In the revised version, we will include this description again and focus on this aspect - to avoid confusion with the Bayesian inference.

*A couple of minor points. On page 6 line 10 you refer to a negatively skew normal distribution of mean zero used to generate errors of the throw for your fault. One would expect such a distribution to have a median value larger than the mean, and there statement that most of the values produced are negative is curious. Please clarify.*

- Thank you for pointing this out, it seems we made a mistake while implementing the skew distribution. As we now plan on using a simpler example model, this distribution will not be used in the revised version.

*Second, the statement of the loss functions from page 11 onward should be clarified. As stated Equations (4)–(7) are identical upto a constant value. I assume that you actually introduce the constants in (5) onward to introduce different degrees of asymmetry. This should be expressed in a rigorous way, indicating the conditions over which the constant is introduced. With some care a parsimonious but rigorous notation could be developed.*

- We intend to include a more detailed description of the loss functions, combined with examples relating to practical decisions, to clarify the definition of the loss functions and to highlight the different options.

---

## Author Comment (AC3) · 13 Jul 2019

**RC2 (Anonymous Referee # 2)**

*This is an interesting and original contribution on simulating uncertainties and their impact on decision making in a structural hydrocarbon trap. Monte Carlo error propagations and Markov Chain Monte Carlo sampling are used to consider the probability of different trap volume models based on stated uncertainties. Loss functions are used to explore potential decision making scenarios for high-risk and low-risk users. The technique has great potential to be used, with addition of more parameters, for hydrocarbon*

[Figure]

*exploration and other geoscience applications.*

- We thank the reviewer for this positive and motivating comment!

*The main strengths of the paper are that it is tightly focused on a specific and important problem, it builds up the methods and results in logical step-wise sections, and it contains useful and clear figures to clarify key inputs/findings. The main weaknesses are that it is written in a fairly inaccessible way, with key elements and some assumptions undefined, and that it does not look beyond the broader applications beyond the specific one used.*

- Thank you for pointing out that the description of the approach is difficult to understand in the current form! We actually had a more extensive description about the design of the loss functions in a previous draft version of the manuscript and we realize now that it will be very beneficial to include it again - with concrete examples and corresponding figures explaining suitable choices. We will include this section again in a revised version of the manuscript.

*There a few ways that it could be made more accessible to a non-specialist in decision analysis (discussed below) and some general editing would shorten the text and increase its clarity.*

*General: For reviewing, it would be much better to have continuous line numbers to refer to rather than numbering resetting back to 1 on each page, also every line rather than every 5th line should be numbered.*

- We agree - this setting has been the default in the LaTeX template that we used - we will change it in the revision for easier editing.

*Lots of instances of unnecessary modifiers e.g. line 10 (P1), line 3-4 (P4), convoluted sentences (e.g. line 2-4, P9) and redundant words (e.g. line 20 P12, line 4, P19) make the text difficult to access. Many of these could be drastically simplified, making the text shorter and more accessible.*

- Thank you very much for your constructive remarks, and especially for highlighting the strengths and weaknesses you identified. You pointed out numerous valid concerns. We will consider them while revising the manuscript to make it more concise and accessible to the reader.

*The word 'actor' is used throughout and in the title. It would be good to define the term in the context used early on. I see that it is commonly used in the decision sciences as a synonym for the more widely-understood term 'decision maker' or 'user', and seems to represent the human element of the process. However, in line 2 (P9) the phrase 'the case of an actor or decision maker' is used, suggesting that the two are different –perhaps actor has a very specific meaning here, which should be made clear. In line11 (P9), for example, the 'actor' would like to know the trap volume. Use of actor in this context seems obscure, and would be better replaced with a common word representing the human element in decision analysis like worker/geologist/company/user. Line15 (P17) reports observations of the behaviour of the modelled actors – reflecting that actors are a non-human element of the modelling (e.g. Fig. 3). Please clarify.*

- We will in particular clarify the definition of 'actor', which we use interchangeably with 'decision maker', earlier in the text. The customized loss function approach is

aimed to include and translate human aspects, such as subjective preferences, so that they are automatically taken into account in the decision-making step applied to modeling. We will also emphasize the link to actual human roles such as experts, geologists and companies.

*Several other key concepts, terms and abbreviations are used throughout, but not all are clearly defined. It would be beneficial to place the work in the framework of such definitions.*

*The use of the hydrocarbon sector as an application for geological modelling is sensible. However, no other options are mentioned. What might other sectors be, and what modelling problems could be solved in the same way (perhaps things like nuclear waste disposal, landslide susceptibility)?*

- The suggestion to broaden the view and consider other sectors for potential application of this approach is very justified, particularly considering cases that involve special risk situations. We will evaluate which parallels can be drawn in other fields, but also how application may differ, and include this to a reasonable extent.

*Throughout, volumetrics are discussed in the sense of 'trap volume' – which is appropriate and supported by the parameters used. However, there is one instance of the term 'reservoir volume', and there is consideration, in section 2.3, of OOIP. This indicates that actual hydrocarbon reserves are a key outcome of the analysis. But converting from a trap volume to a reserve volume of course involves extra parameters such as porosity, net:gross ratios, water saturation etc. These factors and their uncertainties may be as important or more important than the overall trap volume. While it is reasonable that other factors aren't included in the present modelling, it would be beneficial to mention them, justify their omission, and perhaps consider how they might be*

*integrated in a future iteration of the model. I think they are alluded to in Fig 7 (reservoir and recovery parameters) but these should be made explicit, given the geologic focus of the topic.*

- In a previous version of this work, we had included the full OOIP equation and assumed the trap volume (in this simplified example) to essentially replace the net rock volume (A∗ h). We omitted this to keep the manuscript short and put focus on the loss function approach. However, it is a valid point that the OOIP equation should be represented, especially for maintaining the link to real-world geological applications.

*Lines 6-8 (P6). Is there actually an independent uncertainty related to fault offset? Since that parameter can only be inferred indirectly via stratigraphic surface picks, I would argue that there is no additional uncertainty on either the hanging wall or footwall beyond what has already been accounted for by the surface uncertainty (which is the sole observational basis for fault offset). It would be useful to see a short descriptive justification for including this additional uncertainty. The significance of this parameter is clear in Fig. 5, where the smaller probabilities of hanging wall seal and reservoir result (I think) from the additional uncertainty applied. If this extra uncertainty is not justified (I think it is not), then it places the subsequent results in doubt.*

- The fault offset uncertainty was introduced for additional model variability and complexity. We agree that it lacks justification if we assume a more realistic exploration scenario. Furthermore, we now recognized that we implemented a higher degree of complexity than needed and are considering to present a much simpler example (omitting fault offset uncertainty) in a revised version, to not distract from the central propositions of our work.

*Line 14 (P6) OOIP/OOIG – presumably this should be OOIP/OGIP instead?*

*Line 25 (P6). SSF is not defined. I suspect it may be 'shale smear factor' (used without the abbreviation in line 19 (P18)). Please define accordingly.*

*Line 18 (P13). 'Low but positive volumes' – is a negative volume possible/meaningful? If not, simply use 'low volumes'.*

*A few minor typos throughout (e.g. line 24 (P18) should read: an individual's..., line 29 (P18) should read: to what extent...). Please check and amend generally.*

- We overall find your comments and corrections to be highly useful to improve the manuscript.

---

## Author Response (AR1)

**Author's Response**

**RC1 (Anonymous Referee #1)**

*This is an interesting paper, I think that the use of loss functions in this setting is a useful development and one with considerable potential, and I think that the paper is to be welcomed because of this.*

- We thank the reviewer for this motivating content on the topic itself.

*That said, the account of the statistical formulation of the author's model is very unclear and needs considerable improvement.*
*In section 2.2 the authors present their synthetic setting. This can be modelled, by an implicit modelling algorithm, from some set of input values. I assume that these input values are notional values of the z coordinate for a particular contact at each of a set of locations {x, y} in addition to notional values of dip direction and angle. The authors state that they can sample 'deviation values' (I assume they mean errors) for these observations which are then added to the 'known' values. With one major caveat (below), the application of the implicit modelling algorithm to each of a series of 'observations' with stochastic error would generate a series of models of the underlying state of affairs. The caveat is that the authors give no indication of how they specify the joint distribution of errors, i.e. the joint distribution for the errors in z at locations $x_1$, $y_1$ and $x_2$, $y_2$. One can assume only that they treat these errors as independent, but that is most implausible in any geological setting, and could have dramatic consequences for the resulting model output. There are several studies, including some cited in this paper, which have considered the spatial dependence of model errors, and the authors should give this careful consideration in revising their work.*

- The reviewer is right in his criticism of the description of our probabilistic model. We did not focus our attention too much on this aspect, as the probabilistic model is not the central point of the paper (see below).
- In our conceptual case study, errors are indeed treated as independent. The model is primarily aimed at providing an adequate base for generating distributions to apply our loss function to. This is, however, not a limitation - any knowledge about parameter correlations could be included in a concrete application (as the ones mentioned by the reviewer).
  - We clarified this aspect on P11 L18-20

*The authors then attempt to set their work in a Bayesian context. However, they do not make this at all explicit, and it is very difficult to understand or assess what they have done. A Bayesian analysis requires some model, for which some set of parameters are unknown and are treated as random variables to reflect this uncertainty. Prior probability distributions express the modeller's subjective view, prior to examining the data under consideration, about the values of these parameters. Likelihood values can be computed for proposed values of the parameters, and this, possibly in conjunction with some sampling method such as MCMC, is then used to characterize the posterior distribution of the parameters, given the data. Posterior distributions, or samples from them, could then be used, for example, to estimate the expected value of a loss function under a particular decision rule.*

*While the paper starts with some very generalized statements about Bayesian methods (Section 2.4), we are nowhere given a sufficiently clear account of their model system and how it is expressed in Bayesian form. The jump from 2.4 to 2.4.1 is much too large. It is not possible for the reader to extract from section 2.4.1 a sufficiently clear account of what the authors have done to be able to reproduce it in a comparable setting, or to be confident that it has been done correctly. For example, the authors refer to 'layer thickness likelihoods'. I assume that the thickness of a layer is a model parameter (assumed to be the same everywhere in space?). It is not made clear how the likelihood is defined, given a set of notional observations. Are the observations treated as independent? Is this reasonable? I would certainly be concerned if the thicknesses of the separate layers are treated as independent random variables. This does not strike me as implausible and, as assumptions go, it is likely to be a sensitive one in this particular context.*

- We didn't go into too much detail about Bayesian inference, as it is not the central aspect of this work. The loss function approach should equally be applicable in a general probabilistic context, using results from common Monte Carlo simulation.
- We now clearly emphasize the focus on the loss function approach. The manuscript was restructured accordingly: In the methods section, we first describe the concept of loss, loss functions and customization (P3 onwards), with a more in-depth explanation of the customization steps. In addition, we now included a more detailed figure explaining these aspects in a simple 1-D model (P8).
- The significance of Bayesian inference as a method is a secondary aspect of this work. It was used to generate comparable distributions in our synthetic

case study. We use it to update the model with additional information (e.g. combine prior information on interface depth with additional data on layer thickness). We clarified this aspect in the manuscript and provide references to similar previous work (P13-14). However, Bayesian inference is no strict prerequisite for the loss function application.

*In short, I think that the authors need to start again, aiming at the simplest and clearest account they can give of their modelling framework so as to exemplify the loss function concept to best advantage. At present this gets lost in vague and hand-waving description.*

- We took this comment very seriously, as we realized that the main focus of the paper was apparently not clear. As suggested, we streamlined the entire description and focused on the most important methods, with a clearer definition of the terms. Most importantly, we now included a simple 2-input-parameter "1-D model" in the Figure explanation on P8 and we are confident that this figure now helps clarify the main concept and relevance of the paper.

*A couple of minor points. On page 6 line 10 you refer to a negatively skew normal distribution of mean zero used to generate errors of the throw for your fault. One would expect such a distribution to have a median value larger than the mean, and there statement that most of the values produced are negative is curious. Please clarify.*

- Yes, the statement was erroneous and is now corrected (P11 L15).

*Second, the statement of the loss functions from page 11 onward should be clarified. As stated Equations (4)–(7) are identical upto a constant value. I assume that you actually introduce the constants in (5) onward to introduce different degrees of asymmetry. This should be expressed in a rigorous way, indicating the conditions over which the constant is introduced. With some care a parsimonious but rigorous notation could be developed.*

- As mentioned above, the loss function customization is now explained in further detail.

**RC2 (Anonymous Referee #2)**

*This is an interesting and original contribution on simulating uncertainties and their impact on decision making in a structural hydrocarbon trap. Monte Carlo error*

*propagations and Markov Chain Monte Carlo sampling are used to consider the probability of different trap volume models based on stated uncertainties. Loss functions are used to explore potential decision making scenarios for high-risk and low-risk users. The technique has great potential to be used, with addition of more parameters, for hydrocarbon exploration and other geoscience applications.*

- We thank the reviewer for this positive and motivating comment!

*The main strengths of the paper are that it is tightly focused on a specific and important problem, it builds up the methods and results in logical step-wise sections, and it contains useful and clear figures to clarify key inputs/findings. The main weaknesses are that it is written in a fairly inaccessible way, with key elements and some assumptions undefined, and that it does not look beyond the broader applications beyond the specific one used.*

- We thank the reviewer for pointing this aspect out, as well. As described above, we took this criticism very seriously and restructured the entire manuscript, including now a conceptualized step-by-step description of the customization with explanations in a 2-input-parameter "1-D model" in the Figure on page 8.
- We also extended the discussion with additional potential applications of the method.

*There a few ways that it could be made more accessible to a non-specialist in decision analysis (discussed below) and some general editing would shorten the text and increase its clarity.*
*General: For reviewing, it would be much better to have continuous line numbers to refer to rather than numbering resetting back to 1 on each page, also every line rather than every 5th line should be numbered.*

- We agree that this numbering is a bit cumbersome, but we used the standard Copernicus template for our submission and did not find an option for continuous line numbering.

*Lots of instances of unnecessary modifiers e.g. line 10 (P1), line 3-4 (P4), convoluted sentences (e.g. line 2-4, P9) and redundant words (e.g. line 20 P12, line 4, P19) make the text difficult to access. Many of these could be drastically simplified, making the text shorter and more accessible.*

● Thank you very much for your constructive remarks, and especially for highlighting the strengths and weaknesses you identified. You pointed out numerous valid concerns that we considered and tried to improve upon.

*The word 'actor' is used throughout and in the title. It would be good to define the term in the context used early on. I see that it is commonly used in the decision sciences as a synonym for the more widely-understood term 'decision maker' or 'user', and seems to represent the human element of the process. However, in line 2 (P9) the phrase 'the case of an actor or decision maker' is used, suggesting that the two are different –perhaps actor has a very specific meaning here, which should be made clear. In line11 (P9), for example, the 'actor' would like to know the trap volume. Use of actor in this context seems obscure, and would be better replaced with a common word representing the human element in decision analysis like worker/geologist/company/user. Line15 (P17) reports observations of the behaviour of the modelled actors – reflecting that actors are a non-human element of the modelling (e.g. Fig. 3). Please clarify.*

- We now use the word "decision maker" preferably, to avoid confusion. But we also define early on that an estimate-based decision can be referred to as an action a, and thereby a decision maker can also be referred to as actor (P3 L27-28).
- We overall still prefer talking about "decision makers" and "actors", due to the quite abstract nature of our approach. Our example application is not defined specifically enough, to name the exact type of individual, the decision maker could be. Therefore we prefer to clarify it in the introduction (P3 L4), as well as again on P4 L25.
- We also talk about companies as an example on P5 L18-21.
- As the sentence on P17 L15 was easily misunderstood, we dropped it.

*Several other key concepts, terms and abbreviations are used throughout, but not all are clearly defined. It would be beneficial to place the work in the framework of such definitions.*
*The use of the hydrocarbon sector as an application for geological modelling is sensible. However, no other options are mentioned. What might other sectors be, and what modelling problems could be solved in the same way (perhaps things like nuclear waste disposal, landslide susceptibility)?*

- Relevance for other fields is now referred to at the end of the discussion.

*Throughout, volumetrics are discussed in the sense of 'trap volume' – which is appropriate and supported by the parameters used. However, there is one instance of the term 'reservoir volume', and there is consideration, in section 2.3, of OOIP. This indicates that actual hydrocarbon reserves are a key outcome of the analysis. But converting from a trap volume to a reserve volume of course involves extra parameters such as porosity, net:gross ratios, water saturation etc. These factors and their uncertainties may be as important or more important than the overall trap volume. While it is reasonable that other factors aren't included in the present modelling, it would be beneficial to mention them, justify their omission, and perhaps consider how they might be integrated in a future iteration of the model. I think they are alluded to in Fig 7 (reservoir and recovery parameters) but these should be made explicit, given the geologic focus of the topic.*

- Very good point. The OOIP context is now established on P11 23 - P13 L9. OOIP is also mentioned before in the introduction, together with the NPV.

*Lines 6-8 (P6). Is there actually an independent uncertainty related to fault offset? Since that parameter can only be inferred indirectly via stratigraphic surface picks, I would argue that there is no additional uncertainty on either the hanging wall or footwall beyond what has already been accounted for by the surface uncertainty (which is the sole observational basis for fault offset). It would be useful to see a short descriptive justification for including this additional uncertainty. The significance of this parameter is clear in Fig. 5, where the smaller probabilities of hanging wall seal and reservoir result (I think) from the additional uncertainty applied. If this extra uncertainty is not justified (I think it is not), then it places the subsequent results in doubt.*

- *We have now added an according statement that our model is not aimed at complete geological plausibility* was added on P11 L18-20.

*Line 14 (P6) OOIP/OOIG – presumably this should be OOIP/OGIP instead?*
*Line 25 (P6). SSF is not defined. I suspect it may be 'shale smear factor' (used without the abbreviation in line 19 (P18)). Please define accordingly.*
*Line 18 (P13). 'Low but positive volumes' – is a negative volume possible/meaningful? If not, simply use 'low volumes'.*
*A few minor typos throughout (e.g. line 24 (P18) should read: an individual's . . . , line 29 (P18) should read: to what extent . . .). Please check and amend generally.*

- We overall found your comments and corrections highly useful. We named the Shale Smear Factor on P13 L14 and P15 L5, fixed the "low volumes" statement and corrected the typos.

[revised manuscript text omitted]